# PRIME ONCE, THEN REPROGRAM LOCALLY: AN EFFICIENT ALTERNATIVE TO BLACK-BOX MODEL REPROGRAMMING

## ABSTRACT

*Black-box model reprogramming* (BMR) aims to re-purpose black-box pre-trained models (i.e., APIs) for target tasks by learning input patterns (e.g., using Zeroth-Order Optimization (ZOO)) that transform model outputs to match target labels. However, ZOO-based BMR is *inefficient*, requiring *extensive API calls* (could be expensive) and suffering from unstable optimization. More critically, we find this paradigm is becoming ineffective on modern, real-world APIs (e.g., GPT-4o), which can ignore the input perturbations ZOO relies on, leading to negligible performance gains. To address these limitations, we propose **PoRL** (Prime Once, then Reprogram Locally), an alternative strategy that shifts the adaptation task to an amenable local model. PoRL initiates a one-time priming step to transfer knowledge from the service API to a local pre-trained encoder. This single, efficient interaction is then followed by a highly effective white-box model reprogramming directly on the local model. Consequently, all subsequent adaptation and inference rely solely on this local model, *eliminating* further API costs. Experiments demonstrate PoRL's effectiveness where prior methods fail: on GPT-4o, PoRL achieves a +27.8% gain over the zero-shot baseline, a task where ZOO provides no improvement. Broadly, across ten diverse datasets, PoRL outperforms state-of-the-art methods with an average accuracy gain of +2.5% for VLMs and +15.6% for VMs, while reducing API calls by over 99.99%. PoRL thus offers a robust and highly efficient solution for adapting modern black-box models. *Code will be released.*

## 1 INTRODUCTION

In recent years, Model-as-a-Service (MaaS) has emerged as a dominant paradigm for deploying and accessing state-of-the-art (SOTA) machine learning models (Achiam et al., 2023; Brown et al., 2020; Radford et al., 2021; Team et al., 2023). Companies and research institutions often release their large pre-trained models (which we refer to as *service models* in this work) as inference APIs, allowing users to query these powerful models without direct access to model parameters or architecture details. This black-box nature creates a significant challenge for transfer learning (Sun et al., 2024; 2022). Unlike some approaches that may assume access to intermediate features or token embeddings (Ouali et al., 2023; Wang et al., 2024) (see Table 1 for a detailed comparison), this work focuses on the most restrictive black-box setting, where only raw input-prediction access to the service model is available. In this challenging scenario, adapting models to downstream tasks, which typically requires gradient information, becomes particularly difficult. Model Reprogramming (MR) (Cai et al., 2024b; Chen, 2024; Jia et al., 2022; Tsai et al., 2020), known as Visual Reprogramming (VR) or Visual Prompting (VP) in vision domains, has been developed as one solution, enabling adaptation of MaaS models through input-level modifications optimized using zeroth-order optimization (ZOO) (Liu et al., 2018; Spall, 1992; 1997; Tu et al., 2019) based solely on model outputs.

While black-box visual reprogramming approaches such as BAR (Tsai et al., 2020) and BlackVIP (Oh et al., 2023) have demonstrated promising results under this strict input-prediction only setting, they suffer from fundamental limitations. The optimization process, reliant on approximate gradients, is often unstable, and the computational and practical costs are prohibitive (Liu et al., 2020; Zhang et al., 2024). Such methods require numerous API calls not only during the lengthy training phase but also for every inference instance, leading to significant financial burdens and requiring uninterrupted

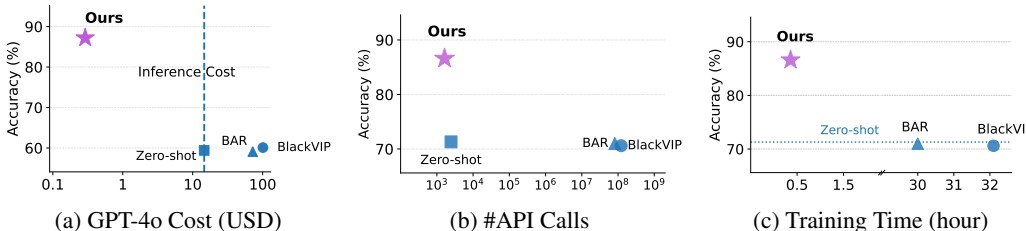

(a) GPT-4o Cost (USD)   (b) #API Calls   (c) Training Time (hour)

Figure 1: Limitations of existing BMR methods. **(a)** On real-world GPT-4o API, ZOO-based methods prove both costly and ineffective, failing to improve upon zero-shot performance. **(b, c)** On CLIP ViT-B/16, these methods require $\sim 10^8$ API calls and over 32 hours of training, yet still underperform our approach, which uses only $\sim 10^3$ API calls and $0.4$ hours.

connectivity to the service model (Oh et al., 2023; Tsai et al., 2020; Wang et al., 2024). More critically, our experiments reveal that this ZOO-based paradigm is becoming ineffective on modern, robust APIs. We find that powerful models like GPT-4o (Hurst et al., 2024) can ignore the noisy input perturbations central to ZOO, resulting in negligible performance gains (Fig. 1a), while on models like LLaVA (Liu et al., 2023), these same perturbations can disrupt delicate vision-language alignment and degrade performance. Furthermore, the substantial resources expended for these SOTA methods do not always translate into commensurate performance enhancements (Fig. 1b and 1c), raising concerns about their efficiency-to-effectiveness ratio (Oh et al., 2023; Yu et al., 2023). These substantial hurdles in efficiency, cost, and now, effectiveness on modern models, form the core motivation for our work. It compels us to ask: *Are there alternative approaches that can offer advantages beyond conventional black-box model reprogramming, particularly in overcoming these practical and economic limitations?*

We propose **PoRL** (Prime Once, then Reprogram Locally), an alternative strategy that directly addresses these challenges (see Fig. 2). Instead of attempting to adapt the service model through continuous black-box optimization, we leverage a locally available pre-trained encoder (e.g., a publicly available one, making the same initial assumption as methods like BlackVIP (Oh et al., 2023) but utilizing it for a fundamentally different purpose). Our key idea is to perform a *one-time* priming step: we query the service API just once with (potentially unlabeled) downstream task data, and use these outputs to prepare a lightweight linear layer on top of the local encoder. After this highly efficient knowledge transfer, the service model API is no longer required for any subsequent adaptation or inference. This two-stage approach, supported by a theoretical analysis connecting priming faithfulness to downstream performance, enables effective white-box visual reprogramming to occur entirely and efficiently on the local model (Cai et al., 2024a; Tsao et al., 2024). For example, on the Flowers102 dataset (as detailed in Fig. 1b and 1c), our approach achieves 86.6% accuracy, outperforming BlackVIP (70.6%), while dramatically reducing API calls from $\sim 10^8$ down to $\sim 10^3$ (>**99.99**% reduction) and slashing computation time from over 30 hours to less than half an hour (>**98.88**% reduction). Crucially, on modern APIs where ZOO-based methods falter, PoRL delivers significant improvements, achieving a +27.8% gain over the zero-shot baseline on GPT-4o. Our approach also translates to completely **cost-free** inference, presenting a truly efficient and economic alternative to ZOO-based BMR.

We demonstrate the significant efficacy of our method considering a broad range of service models—including Vision-Language Models (VLMs) such as CLIP (Radford et al., 2021) with a ViT-B/16 backbone, and standard Vision Models (VMs) like ViT-B/16 (Dosovitskiy et al., 2020) and ResNet101 (He et al., 2016)—evaluated over ten diverse datasets. Through these comprehensive experiments, we demonstrate that our approach consistently outperforms SOTA ZOO-based BMR methods while dramatically reducing computational requirements and associated costs. These findings have important implications for the broader field of transfer learning in increasingly API-centric machine learning ecosystems, suggesting a more effective path forward that balances performance, efficiency, and practicality. The main contributions of our work are summarized as follows:

1. We identify a critical limitation of ZOO-based reprogramming on modern, robust APIs and present an effective alternative that shifts adaptation to a local model after a minimal, one-time interaction with the service API.

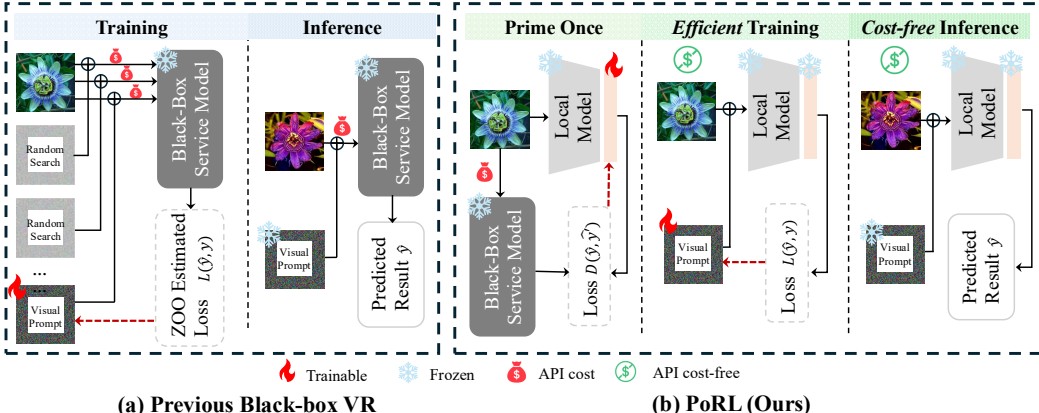

Figure 2: Comparison between previous Black-box Visual Reprogramming (VR) methods and our proposed **PoRL** approach. (a) Previous black-box methods utilize Zeroth-Order Optimization (ZOO) and require continuous API access (and associated costs) during both training and inference. (b) Our method performs a one-time priming from the Service model to a Local encoder, then enables efficient gradient-aware VR locally, eliminating all API costs during inference.

2. We introduce **PoRL** (Prime Once, then Reprogram Locally), a novel framework supported by theoretical analysis that combines a one-time priming step with efficient local model reprogramming, uniquely enabling cost-free inference without further API dependency.

3. Through extensive experiments, PoRL achieves superior performance, including significant gains on real-world APIs (e.g. GPT-4o and Clarifai) where prior methods fail, alongside average accuracy improvements of $+2.5\%$ for VLM and $+15.6\%$ for VM over BlackVIP, while reducing API calls by $> 99.99\%$ and training time by $> 98\%$.

## 2 RELATED WORK

**Black-box Tuning for Model-as-a-Service (MaaS).** Black-box tuning for MaaS has been explored across various modalities, with approaches for Large Language Models (LLMs) (Sun et al., 2024; 2022) and methods for VLMs that modify inputs at the token level while requiring access to text embeddings, placing them outside the strictest black-box definition (Guo et al., 2023; Ouali et al., 2023; Wang et al., 2024; Yu et al., 2023), as detailed in Table 1. In the vision domain, approaches like BlackVIP (Oh et al., 2023) and BAR (Tsai et al., 2020) employ ZOO to reprogram models by modifying visual inputs. While applicable to both VLMs and VMs, these ZOO-based approaches are known for optimization instability and high API call costs during training and inference (Liu et al., 2020; Oh et al., 2023; Tsai et al., 2020; Zhang et al., 2024). More critically, their reliance on input perturbations is emerging as a key limitation in the era of modern APIs, e.g., powerful service models like GPT-4o can be robust enough to simply ignore these perturbations, rendering the costly ZOO adaptation process ineffective (Fig. 1a). In contrast, PoRL shifts the adaptation to a local model via a one-time priming step, enabling efficient and stable reprogramming without further API dependency.

**Model Reprogramming (MR).** MR (Bahng et al., 2022; Chen, 2024; Jia et al., 2022; Zhang et al., 2022), termed Visual Reprogramming (VR) or Visual Prompting (VP) in vision domains, encompasses two main strategies. The first, token-based prompting, is tailored for transformer architectures like Vision Transformers (ViT) (Bahng et al., 2022; Jia et al., 2022; Zeng et al., 2024; Zhou et al., 2022a;b). These methods inject learnable tokens and require access to the model's embedding space or intermediate layers, rendering them unsuitable for strict black-box settings or non-transformers (Guo et al., 2023; Ouali et al., 2023; Wang et al., 2024; Yu et al., 2023). The second category, input-level prompting (Cai et al., 2024a; Chen et al., 2023; Oh et al., 2023; Tsai et al., 2020; Tsao et al., 2024), directly modifies input images, offering model-agnostic flexibility with only input-output access. Both approaches can be optimized using gradient-based methods (white-box) or ZOO techniques (black-box) (Oh et al., 2023; Tsai et al., 2020; Wang et al., 2024). However, ZOO-based methods suffer from significant efficiency and stability issues, which our work addresses.

Table 1: Comparison of black-box tuning methods for Model-as-a-Service (MaaS). "MR" indicates if the method is Model Reprogramming based (operating on raw inputs). "Input Acc." and "Output Acc." denote required access to service model inputs (raw Input vs. Embedding) and outputs (Predictions vs. Feats.). "VM"/"VLM" support, "Prompt" type (L: Language, V: Visual), and "Enc. Flex." (encoder flexibility) are also shown. Our method, PoRL, uniquely offers inference-time API cost-free adaptation ("Free Inf.") for both VMs and VLMs under the strictest black-box constraints.

| Access | Method | MR | Input Acc. | Output Acc. | VM | VLM | Prompt | Enc. Flex. | Free Inf. |
|--------|--------|----|-----------|-------------|----|-----|--------|-----------|-----------|
| | LFA (Ouali et al., 2023) | ✗ | Input | Feats. | ✗ | ✓ | – | ✓ | ✗ |
| | CBBT (Guo et al., 2023) | ✗ | Embedding | Feats. | ✗ | ✓ | L | ✓ | ✗ |
| | LaViP (Kuna. et al., 2024) | ✓ | Input | Feats. | ✓ | ✓ | V | ✓ | ✗ |
| | BPT-VLM (Yu et al., 2023) | ✗ | Embedding | Preds. | ✗ | ✓ | L&V | ViT Only | ✗ |
| | CraFT (Wang et al., 2024) | ✗ | Embedding | Preds. | ✗ | ✓ | L | ✓ | ✗ |
| | BAR (Tsai et al., 2020) | ✓ | Input | Preds. | ✓ | ✓ | V | ✓ | ✗ |
| | BlackVIP (Oh et al., 2023) | ✓ | Input | Preds. | ✓ | ✓ | V | ✓ | ✗ |
| | LLM-Opt (Liu et al., 2024) | ✗ | Input | Preds. | ✗ | ✓ | L | ✓ | ✗ |
| | **PoRL (Ours)** | ✓ | Input | Preds. | ✓ | ✓ | V | ✓ | ✓ |

## 3 PROBLEM SETTING

**Balck-box Model Reprogramming.** MR adapts pre-trained models to solve new tasks without modifying their parameters. Given a pre-trained model $\mathcal{F}_S : \mathcal{X}^S \mapsto \mathbb{R}^{K^S}$ where $K^S$ is the number of source classes, the key concept is to introduce a learnable prompt $\mathbf{P}$ that transforms inputs from the downstream task domain $\mathcal{X}^T$ to the source domain $\mathcal{X}^S$ through an input transformation function $g_{\text{in}}(\cdot|\mathbf{P})$. Additionally, a label mapping function $g_{\text{out}} : \mathbb{R}^{K^S} \mapsto \mathbb{R}^{K^T}$ is employed to bridge the source (with $K^S$ classes) and target (with $K^T$ classes) label spaces. If the source and downstream tasks have the same label space, $g_{\text{out}}$ becomes an identity mapping (e.g., in VLMs). Otherwise, for VMs, a gradient-free label mapping is established following existing works (Cai et al., 2024a; Chen et al., 2023; Tsai et al., 2020) (Detailed in Appendix A.2). Importantly, only the input transformation $g_{\text{in}}$ contains trainable parameters updated through gradient descent, while the label mapping $g_{\text{out}}$ is computed and updated during training but does not introduce additional trainable parameters.

In MaaS settings, powerful pre-trained models are accessible only through APIs without access to their parameters or internal representations. We denote such a black-box Service API model as $\mathcal{F}_S : \mathcal{X}^S \mapsto \mathbb{R}^{K^S}$, which accepts inputs from domain $\mathcal{X}^S$ and produces output predictions. Adhering to the strictest black-box assumption established in prior work (Tsai et al., 2020), we consider the scenario where the API returns only prediction probabilities rather than logits or embeddings, representing the most constrained access scenario (see Table 1). When adapting MaaS models to downstream tasks, the central challenge is to leverage these powerful models in a computationally and economically efficient manner. This requires developing methods that can optimize task-specific visual prompts using only the limited input-output responses from the API while minimizing computational costs and API usage during both training and inference.

**Existing BMR Approaches.** Existing ZOO-based BMR methods like BAR (Tsai et al., 2020) employs random gradient-free (RGF) optimization (Liu et al., 2018; Tu et al., 2019) to estimate the gradient of the loss function $\ell$ with respect to prompt parameters $\mathbf{P}$: $\nabla\ell(\mathbf{P}) \approx \frac{d}{q}\sum_{i=1}^{q}\frac{\ell(\mathbf{P}+\mu u_i)-\ell(\mathbf{P})}{\mu}u_i$, where $d$ is the parameter dimension, $q$ is the number of random directions, $\mu$ is a small constant, and $u_i$ are random unit vectors. BlackVIP (Oh et al., 2023) introduces the Simultaneous Perturbation Stochastic Approximation with Gradient Correction (SPSA-GC) (Spall, 1992; 1997) to improve gradient estimation: $\nabla\ell(\mathbf{P}) \approx \frac{\ell(\mathbf{P}+c\Delta)-\ell(\mathbf{P}-c\Delta)}{2c}\Delta^{-1}$, where $c$ is a small constant and $\Delta$ is a random perturbation vector. BlackVIP also incorporates a Coordinator network for input-dependent prompt generation using a local pre-trained encoder (e.g., ViT-B/16). While these methods have shown promising results, they require multiple API calls per update and during inference, leading to increased computational and financial costs. Additionally, their zeroth-order gradient approximation can result in less stable optimization compared to exact gradients (i.e., first-order optimization) (Liu et al., 2020; Oh et al., 2023; Zhang et al., 2024). The challenge remains to develop more efficient approaches that can better leverage available local encoders while minimizing API usage.

## 4 PoRL: Prime Once, then Reprogram Locally

**Prime Once.** In the first stage, we utilize a locally available encoder $\mathcal{F}_L$ (the same assumption as BlackVIP (Oh et al., 2023)) and prime it using the black-box service model $\mathcal{F}_S$. For each image $x_i$ from the target domain dataset $\mathcal{D}^T = \{x_i\}_{i=1}^N$, we query the service model to obtain prediction probabilities $p_S(x_i) = \mathcal{F}_S(x_i)$. It is important to note that this is the *only time* we interact with the API, and after this single round of querying, the API is *no longer needed* for the rest of the process. We use the local encoder $\mathcal{F}_L$ (outputing $d_{\text{enc}}$-dim features) as a fixed feature extractor and add a trainable linear layer with parameters $\theta \in \mathbb{R}^{K^S \times (d_{\text{enc}}+1)}$ on top of it. Note that we abuse the notation $\mathcal{F}_L$ to refer to both the local encoder alone and the local encoder combined with the learned linear layer, depending on the context. Only the linear layer parameters are updated during priming:

$$\theta^* = \arg\min_\theta \sum_{i=1}^N \mathcal{L}_{\text{P}}(\mathcal{F}_L(x_i; \theta), p_S(x_i)). \tag{1}$$

Our priming objective, inspired by the loss function from knowledge distillation (Hinton et al., 2015; Stanton et al., 2021), is defined as: $\mathcal{L}_{\text{P}}(p_L, p_S) := -\sum_{j=1}^{K^S} p_{S,j} \log p_{L,j}$, where $p_S/p_L$ are the output probabilities of the same image from the service/local models, respectively. This loss is equivalent to the Kullback-Leibler divergence $\text{KL}(p_S||p_L)$.

For both VMs and VLMs, we freeze the local encoder's weights and only train the added linear layer. The key difference between VM and VLM adaptation is in how the outputs are handled: for VMs, the priming occurs in the service model label probability space, while for VLMs, the outputs are naturally aligned with the downstream task label space through the text encoder. This approach establishes a fair comparison with BlackVIP (Oh et al., 2023), as both methods only require a local encoder. Moreover, this opens up the potential to utilize extra **unlabeled** data from the downstream distribution, a significant advantage unavailable to other methods. (see Fig. 3d).

**Remark** (Priming vs. Knowledge Distillation). Our priming stage is mechanistically similar to knowledge distillation (Ba & Caruana, 2014; Chen et al., 2017; Hinton et al., 2015; Hsu et al., 2021; Ji & Zhu, 2020; Phuong & Lampert, 2019) but differs fundamentally in its objective and applicability. Whereas traditional distillation aims to create a final, high-performance student model—typically requiring a shared label space—our priming is solely a preparatory step to make the local model more amenable to subsequent reprogramming. This preparatory focus allows priming to operate even when the service and downstream label spaces are disjoint (e.g., using `Flowers` data in the `ImageNet` label space), providing crucial flexibility for black-box model adaptation.

**Efficient VR Training.** After the priming, we proceed to the second stage of PoRL: learning an effective visual prompt for the downstream task using gradient-based optimization. Unlike black-box approaches that rely on approximate gradients, we can leverage exact gradient information since our local encoder with the learned linear layer is fully accessible. We define a learnable visual prompt $\mathbf{P}$ and an input transformation function $g_{\text{in}}$ similar to black-box methods, along with a label mapping function $g_{\text{out}}$ (and becomes an identity mapping for VLMs). We optimize the prompt using gradient descent on our primed local encoder by minimizing the cross-entropy loss:

$$\mathbf{P}^* = \arg\min_{\mathbf{P}} \mathbb{E}_{(x,y)\sim\mathcal{D}^T}[\ell(g_{\text{out}}(\mathcal{F}_L(g_{\text{in}}(x, \mathbf{P}); \theta^*)), y)]. \tag{2}$$

This gradient-based optimization on local model enables more stable, efficient, and rapid learning of the visual prompt compared to ZOO-based methods, resulting in better and faster adaptation to the downstream task while maintaining the benefits of input-level modifications.

**Cost-free Inference.** For inference on test examples from the downstream task, we apply the learned visual prompt to the input and process it through the local encoder $\hat{y} = g_{\text{out}}(\mathcal{F}_L(g_{\text{in}}(x, \mathbf{P}^*); \theta^*))$. This approach eliminates API dependency during inference, enabling truly cost-free deployment. By removing the need for constant connectivity to service models, PoRL becomes particularly valuable in practical scenarios like remote field operations or edge devices with intermittent network access.

**Model Selection Strategy.** For applications where maximizing performance is the top priority, we also propose a simple variant named **PoRL-MS** (Model Selection). This approach leverages the initial priming stage as a low-cost diagnostic tool, as it yields both the zero-shot performance of the service API and the potential performance of the primed local model. PoRL-MS then automatically selects the optimal inference path: if the local model's potential meets the required performance threshold, defined as the zero-shot performance minus a tolerance $\tau$, it proceeds with the cost-free

local model; otherwise, it defaults to using the more powerful zero-shot API. This turns PoRL into a more complete framework that not only enables efficient adaptation but also guides practitioners to the optimal cost-performance trade-off for each task.

**Theoretical Insight.** To formally understand PoRL's efficacy, we provide a theoretical analysis (detailed in Appendix C) that bounds the performance of our primed local model, $\mathcal{F}_L$, relative to the service model, $\mathcal{F}_S$. This analysis hinges on two key assumptions: *Service Model Superiority* (Assumption 1), which reasonably posits that $\mathcal{F}_S$ is more powerful or better aligned with the downstream task data $\mathcal{D}^T$ than $\mathcal{F}_L$ initially, and $\epsilon$-*Faithful Priming* (Assumption 2 and Definition 1). The latter states our one-time priming effectively aligns the output logit distributions of $\mathcal{F}_L$ and $\mathcal{F}_S$, with their expected $L_1$ difference bounded by a small $\epsilon$, indicating a successful knowledge transfer. While our theoretical analysis assumes *logits* for tractability, our ablation studies (Fig. 3c) confirm that the performance of our practical, probability-only method is unaffected.

Leveraging these assumptions and the Lipschitz continuity of the cross-entropy loss (Lemma 1), Theorem 3 establishes a crucial performance bound. It shows that the risk of the optimally reprogrammed local model, $\mathcal{R}_L(\mathcal{D}^T, \mathbf{P}^*)$, can closely approach that of an optimally reprogrammed service model, $\mathcal{R}_S(\mathcal{D}^T, \mathbf{Q}^*)$, differing by at most $\epsilon$:

$$\mathcal{R}_L(\mathcal{D}^T, \mathbf{P}^*) - \epsilon \le \mathcal{R}_S(\mathcal{D}^T, \mathbf{Q}^*) \le \mathcal{R}_L(\mathcal{D}^T, \mathbf{P}^*). \tag{3}$$

This result offers a key insight: traditional ZOO-based methods expend considerable resources attempting to directly optimize for a low $\mathcal{R}_S(\mathcal{D}^T, \mathbf{Q}^*)$ via unstable, query-heavy processes on $\mathcal{F}_S$. PoRL, however, transforms this challenge. By first achieving a faithful priming (small $\epsilon$), the problem shifts to optimizing $\mathcal{R}_L(\mathcal{D}^T, \mathbf{P}^*)$ on the local, white-box model $\mathcal{F}_L$. This local optimization using efficient First-Order Optimization (FOO) is more stable and drastically reduces API calls and computation, thus providing a more practical path to leveraging the service model's capabilities.

## 5 EXPERIMENTS

**Datasets and Models.** We evaluate PoRL on ten diverse visual recognition datasets widely used in previous works (Oh et al., 2023; Cai et al., 2024a), spanning various domains: fine-grained categorization (Flowers102 (Nilsback & Zisserman, 2008), StanfordCars (Krause et al., 2013)), texture recognition (DTD (Cimpoi et al., 2014)), action recognition (UCF101 (Soomro et al., 2012)), food classification (Food101 (Bossard et al., 2014)), traffic sign recognition (GTSRB (Houben et al., 2013)), satellite imagery (EuroSAT (Helber et al., 2019)), animal recognition (OxfordPets (Parkhi et al., 2012)), scene classification (SUN397 (Xiao et al., 2010)), and digit recognition (SVHN (Netzer et al., 2011)). We compare our approach with SOTA black-box VR methods: BAR (Tsai et al., 2020) and BlackVIP (Oh et al., 2023). For the VLM experiments, we additionally compare against LLM-Opt (Liu et al., 2024), a recent baseline applicable only to VLMs. For VLMs, we use CLIP ViT-B/16 (Radford et al., 2021) as the service model and ViT-B/16 as the local encoder in a 16-shot learning setting, following the same configuration as BlackVIP (Oh et al., 2023) due to VLMs' strong generalization capabilities. We also include zero-shot CLIP performance as an important baseline to measure adaptation gains. For VMs, we employ pre-trained ViT-B/16 (Dosovitskiy et al., 2020) and ResNet101 (RN101) (He et al., 2016) as service models with ViT-B/32 and ResNet50 (RN50) as local encoders, adopting the Full-shot setting following BAR (Tsai et al., 2020) since VMs are not as generalizable as VLMs. For all settings using VMs as a service, we employ Bayesian-guided Label Mapping (BLM) (Cai et al., 2024a) as $g_{\text{out}}$. In addition, we further evaluate our methods on the Large Vision-Language Model LLaVA (Liu et al., 2023), the real-world API GPT-4o (Hurst et al., 2024), and the commercial API service Clarifai. Additional details can be found in Appendix B.

**Implementation Details and Evaluation Metrics.** For the priming stage in PoRL, we use the Adam optimizer (Kingma & Ba, 2014) with a learning rate of 0.001. For local encoder VR, we follow the implementation in (Cai et al., 2024a), using a padding-based visual prompt approach and the Adam optimizer with a learning rate of 0.01. For baselines, we use the officially released implementations with their recommended hyperparameter settings. All experiments are conducted *three* rounds on a single NVIDIA RTX 6000 Ada GPU. We evaluate each method using three key metrics to assess practical deployment value: (1) Average accuracy across three runs on downstream tasks (**%**), (2) Total training and inference API calls required in millions (**#API (M)**), and (3) Total wall-clock training time (**Time (h)**) across all datasets. For experiments on real-world APIs, we also report the total financial cost in USD ($) for both training and inference.

Table 2: Accuracy and Efficiency comparison on ten diverse visual recognition datasets using CLIP ViT-B/16 (Service) and ViT-B/16 (Local) in a 16-shot setting. PoRL achieves superior average accuracy with a drastic reduction in resources (99.99% fewer API calls and 98% less training time). For this and all subsequent tables, Gray denotes white-box Service VR results. #API (M): total API calls in millions (training and inference); Time (h): total training hours on an RTX 6000 Ada GPU.

| Method | Flowers | DTD | UCF | Food | GTSRB | EuroSAT | Pets | Cars | SUN | SVHN | Avg. | #API (M) | Time (h) |
|---|---|---|---|---|---|---|---|---|---|---|---|---|---|
| VR (white-box) | 86.8 | 62.0 | 74.0 | 81.6 | 65.1 | 90.9 | 90.7 | 66.2 | 66.7 | 60.2 | 74.4 | 16.82 | 9.8 |
| Zero-shot | 71.3 | 43.9 | 66.9 | **85.9** | 21.0 | 47.9 | **89.1** | 65.2 | 62.6 | 17.9 | 57.2 | 0.12 | 0 |
| BAR | 71.0 | 46.8 | 64.2 | 84.4 | 21.5 | 77.3 | 88.4 | 63.0 | 62.4 | 34.6 | 61.4 | 612.84 | 185.6 |
| BlackVIP | 70.6 | 45.3 | **68.7** | **85.9** | 21.3 | 73.3 | **89.1** | 65.4 | 64.5 | 44.4 | 62.9 | 754.20 | 197.5 |
| LLM-Opt | 67.6 | 45.0 | 59.9 | 78.5 | 21.2 | 48.0 | 87.7 | 56.2 | 60.3 | 20.2 | 54.5 | 5011.32 | 5.1 |
| **PoRL** | **86.6** | **48.2** | 67.1 | 68.8 | 39.4 | 85.7 | 88.9 | 43.2 | 62.8 | 63.2 | 65.4 | **0.02** | 3.7 |
| **PoRL-MS** | **86.6** | **48.2** | 67.1 | 85.9 | 39.4 | 85.7 | 88.9 | 65.2 | 62.8 | 63.2 | **69.3** | **0.06** | 3.7 |

Table 3: Accuracy and Efficiency comparison using ViT-B/16 (Service) in Full-shot setting.

| Method | Flowers | DTD | UCF | Food | GTSRB | EuroSAT | Pets | Cars | SUN | SVHN | Avg. | #API (M) | Time (h) |
|---|---|---|---|---|---|---|---|---|---|---|---|---|---|
| VR (white-box) | 69.6 | 52.1 | 49.7 | 38.7 | 60.6 | 96.7 | 77.1 | 6.2 | 34.0 | 83.3 | 56.8 | 43.4 | 39.6 |
| BAR | 14.8 | 24.3 | 29.5 | 12.2 | 14.1 | 43.1 | 24.9 | 1.0 | 10.6 | 29.6 | 20.4 | 1,724.2 | 668.2 |
| BlackVIP (RN50) | 15.2 | 42.3 | 34.0 | 14.5 | 13.9 | 67.3 | 63.6 | 3.7 | 22.6 | 30.8 | 30.8 | 2,586.2 | 686.9 |
| **PoRL (RN50)** | **43.5** | **43.4** | **39.4** | **25.9** | **48.7** | **84.3** | **73.9** | **5.8** | **19.9** | **73.7** | **45.9** | **0.2** | **8.7** |
| BlackVIP (ViT-B/32) | 27.4 | 44.1 | 36.6 | 18.5 | 38.4 | 59.6 | 65.4 | 3.9 | 23.7 | 30.3 | 34.8 | 2,586.2 | 757.6 |
| **PoRL (ViT-B/32)** | **53.6** | **46.9** | **41.4** | **27.3** | **58.8** | **95.6** | **65.0** | **4.7** | **26.9** | **83.5** | **50.4** | **0.2** | **22.4** |

## 5.1 EXPERIMENTAL RESULTS

**Results on Vision-Language Model (VLMs).** Following BlackVIP (Oh et al., 2023), we evaluate our approach using a CLIP ViT-B/16 service model and a ViT-B/16 local encoder, with results presented in Table 2. Our method, **PoRL**, achieves a 65.4% average accuracy across ten diverse datasets, outperforming both BAR (61.4%) and BlackVIP (62.9%). We also compare against LLM-Opt (Liu et al., 2024), a recent method that uses GPT-4 to optimize text prompts; despite its extremely high API costs for both CLIP (5011.3M API calls) and the LLM optimizer ($\sim$\$110 USD), it underperforms the zero-shot baseline and is only applicable to VLMs. This superior performance by PoRL is achieved with a drastic reduction in computational requirements: only 0.02M API calls for the initial priming, compared to 612.84M for BAR and 754.20M for BlackVIP (a reduction of $>$ 99.99%). Furthermore, inferring all datasets' test sets (e.g., zero-shot) requires 0.12M API calls; our method uses only approximately 1/6 of these calls for priming and then requires no API calls during inference, while still outperforming zero-shot by $+8.2$% on average. Training time is also significantly reduced to only 3.7 hours, compared to 185.6 and 197.5 hours for BAR and BlackVIP respectively (a reduction of $>$ 98%). These efficiency gains are particularly notable on challenging datasets such as SVHN (63.2% vs. 44.4% for BlackVIP and 34.6% for BAR) and EuroSAT (85.7% vs. 73.3% for BlackVIP and 77.3% for BAR). Our performance on Flowers102 is also strong at 86.6%, substantially outperforming BlackVIP (70.6%) and BAR (71.0%), and closely approaching the white-box VR performance (86.8%) without direct use of the service model.

On certain complex datasets, notably Food101 and Cars, we observe that all evaluated reprogramming methods, including our own (with a failure case analysis provided in Appendix D.7), struggle to outperform the strong zero-shot CLIP baseline. This suggests a potential inherent limitation of input-level visual reprogramming for these specific domains. This hypothesis is strongly supported by the results in Table 2, where a fully white-box VR approach is shown to be ineffective, even causing a 4.3% performance *decrease* on Food101 compared to the zero-shot baseline. This insight motivates our **PoRL-MS** variant, which functions as an intelligent model selection strategy. By using the priming stage to identify these challenging datasets and defaulting to the zero-shot API, PoRL-MS boosts the average accuracy to a new state-of-the-art 69.3% (Table 2), effectively navigating the cost-performance trade-off by applying reprogramming only where it is beneficial.

**Results on Vision Models (VMs).** We further validate our approach using ViT-B/16 as the black-box service model, paired with ViT-B/32 and RN50 local encoders (details in Table 3; results using an RN101 service model are in Appendix Table 8). Our method consistently outperforms BAR and BlackVIP across all configurations. As shown in the main results, we achieve an average accuracy of up to 50.4 (with a ViT-B/32 local encoder), compared to significantly lower accuracy for BAR (20.4%) and BlackVIP (34.8%). This superior performance comes with drastically reduced resource

Table 4: Performance and Cost comparison on modern MLLMs and Real-world commercial APIs.

| Method | LLaVA | | | | GPT-4o | | | | Clarifai | | | |
|---|---|---|---|---|---|---|---|---|---|---|---|---|
| | Acc. ↑ | Train | Infer. | Total (#) ↓ | Acc. ↑ | Train | Infer. | Total ($) ↓ | Acc. ↑ | Train | Infer. | Total ($) ↓ |
| White-box | 91.3 | – | – | – | – | – | – | – | – | – | – | – |
| Zero-shot | 40.1 | – | 8100 | 8100 | 59.4 | – | 14.6 | 14.6 | – | – | – | – |
| BAR | 34.1 | $\sim 10^6$ | 8100 | $\sim 10^6$ | 59.1 | 57.6 | 14.6 | 72.2 | 68.1 | 38.4 | 9.7 | 48.1 |
| BlackVIP | 39.4 | $\sim 10^7$ | 8100 | $\sim 10^7$ | 60.1 | 86.4 | 14.6 | 101.0 | 72.1 | 57.6 | 9.7 | 67.3 |
| **PoRL** | **73.1** | 160 | 0 | **160** | **87.2** | 0.3 | 0 | **0.3** | **83.2** | 0.2 | 0 | **0.2** |

requirements; in the full-shot setting, our method uses less than 0.02% of the API calls and only 3-7% of the computation time required by competitors like BlackVIP, which can demand over 2500M API calls and 750 hours of training.

To further analyze the effectiveness of our priming stage, we compare PoRL against a direct white-box VR baseline performed on the local model. Surprisingly, simply reprogramming the local encoder directly already outperforms the complex ZOO-based BlackVIP (e.g., 45.3% vs. 34.8% for the ViT-B/32 local model). Our priming process then further elevates this strong baseline by a significant margin (e.g., an additional +5.1% to 50.4%). Moreover, our approach often exhibits synergistic effects, surpassing both the local VR baseline and the estimated white-box performance of the service model on several datasets. For instance, with an RN101 service and ViT-B/32 local model, on Flowers102 our method achieves 51.3% (vs. 42.3% service white-box vs. 38.7% local VR). This demonstrates our method's potential to effectively combine knowledge from both models for enhanced reprogramming outcomes with high efficiency.

**Results on MLLMs and Real-World APIs.** To validate PoRL's practical effectiveness and efficiency, we conducted a targeted evaluation on modern services using the EuroSAT 16-shot benchmark. We test PoRL against baselines on three distinct models: the open-source VLM LLaVA, the proprietary VLM GPT-4o, and the commercial VM Clarifai. For the commercial APIs, we report the real-world cost in USD, for which white-box baselines are unavailable. As shown in Table 4, PoRL demonstrates substantial gains in both accuracy and efficiency, confirming its value in real-world settings.

PoRL's strong performance stems from its two-stage design, which circumvents challenges faced by ZOO-based methods on modern models. On LLaVA, for example, PoRL avoids using input perturbations that appear to disrupt the model's delicate vision-language alignment, a problem that causes BAR and BlackVIP's performance to drop below the Zero-shot baseline. Similarly, on the robust GPT-4o, where ZOO-based methods struggle to find effective perturbations, PoRL's local reprogramming remains highly effective, achieving a remarkable +27.8% gain over Zero-shot. On the commercial Clarifai API—a closed-vocabulary model where zero-shot is not an option—PoRL again achieves the highest accuracy (83.2%) at a fraction of the cost ($0.2), making it over *300x cheaper* than BlackVIP ($67.3). These results confirm that PoRL is a robust, effective, and economically viable solution for adapting modern black-box models. More details in the Appendix D.8.

## 5.2 ABLATION STUDY

We present ablation studies in this section. All experiments use EuroSAT with 16 shots per class, CLIP ViT-B/16 (Service) and ViT-B/16 (Local), unless specified otherwise.

**Backbone Versatility.** We examine the effect of varying service model backbones and local encoder choices to demonstrate our approach's versatility when the service model is VLM.

The results in Table 5 indicate that our method, whether using a ViT-B/16 or an RN50 local encoder, consistently outperforms baselines like BAR and BlackVIP across diverse CLIP backbones (RN50, RN101, ViT-B/32, and ViT-B/16) used in VLM contexts. For instance, averaged across these service backbones, our method with a ViT-B/16 local encoder achieves an average accuracy of 82.9%, substantially higher than BlackVIP with the same local encoder (60.2%).

Table 5: Ablation study across backbones.

| Method | Service Backbone | | | | |
|---|---|---|---|---|---|
| | RN50 | RN101 | ViT-B/32 | ViT-B/16 | Avg. |
| ZS | 37.5 | 32.6 | 45.2 | 47.9 | 40.8 |
| BAR | 26.9 | 33.5 | 70.3 | 77.2 | 52.0 |
| BlackVIP (RN50) | 51.3 | 50.8 | 62.9 | 68.5 | 58.4 |
| **PoRL** (RN50) | **62.2** | **68.5** | **63.2** | **78.6** | **68.1** |
| BlackVIP (ViT-B/16) | 48.4 | 51.3 | 67.9 | 73.3 | 60.2 |
| **PoRL** (ViT-B/16) | **81.8** | **81.9** | **82.0** | **85.7** | **82.9** |

This underscores the robustness of our priming and local reprogramming strategy across different architectural choices.

**Component Analysis.** In Table 6, we dissect the individual contributions of Priming Stage (PS) and

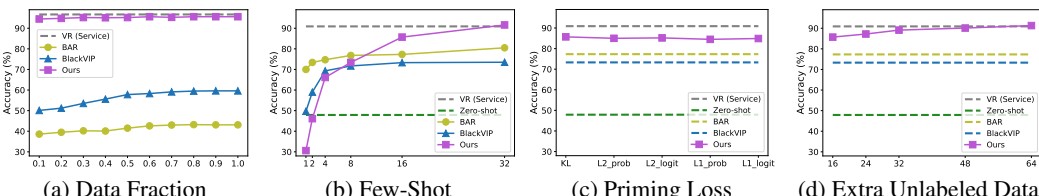

| (a) Data Fraction | (b) Few-Shot | (c) Priming Loss | (d) Extra Unlabeled Data |

Figure 3: Ablation studies for PoRL on the EuroSAT dataset: impact of (a) training data fraction (for VMs), (b) few-shot sample size (for VLMs), (c) priming loss function, and (d) amount of extra unlabeled data for priming on accuracy (%). Default setting involves 16-shot learning, CLIP ViT-B/16 Service, and ViT-B/16 Local encoder unless otherwise indicated by the subplot analysis.

local Visual Reprogramming (VR). Simply using the priming stage to prepare a linear layer on top of the local encoder from the service model yields an accuracy of 45.6%, lower than the service model's zero-shot performance (47.9%) on EuroSAT. Pre-priming local adaptation methods show varied success: Local VR (with an additional pre-trained linear layer on the source domain, as discussed in Section 5.1) achieves 70.6%, Local Linear Probing (LP) reaches 73.8%, and their combination (Local VR+LP) results

Table 6: Component analysis.

| Method | PS | VR | LP | Acc. |
|---|---|---|---|---|
| Zero-shot | – | – | – | 47.9 |
| PS | ✓ | – | – | 45.6 |
| Local VR | – | ✓ | – | 70.6 |
| Local LP | – | – | ✓ | 73.8 |
| Local VR+LP | – | ✓ | ✓ | 80.1 |
| **PoRL (Ours)** | ✓ | ✓ | – | **85.7** |

in 80.1% accuracy. However, our full approach, which first employs PS and then applies VR to the primed local model, achieves the highest accuracy of 85.7%, demonstrating that both components are crucial and contribute synergistically to the final performance.

**Impact of Training Data Size.** The impact of training data size is investigated for both VMs and VLMs. For VMs, Fig. 3a shows our method consistently outperforms BAR and BlackVIP, maintaining significant performance even when data is scarce. Additional VM evaluation in the 16-shot setting is provided in Appendix D.1. For VLMs, varying the number of samples per class in a few-shot setting is presented in Fig. 3b. While at very low shot counts (e.g., $< 8$ shots), the performance of our method can be comparable or slightly underperform some baselines due to suboptimal priming and local VR training from extreme data scarcity, it stabilizes quickly. From 8 shots onwards, our method consistently and significantly outperforms baselines, and with more data (e.g., 32-shots), its performance can be further improved to levels comparable to white-box service model VR.

**Priming Strategies.** We explore different priming loss functions and the utility of extra unlabeled data, with results in Fig. 3c and Fig. 3d respectively. Among various priming losses (KL divergence, L1/L2 losses on probabilities or logits), KL divergence, used in our main experiments, provides strong and stable performance. While we operate under the strict black-box assumption (probabilities only), logits-based losses are included for broader comparative analysis. Furthermore, a key benefit of priming is its lack of requirement for extra labeled data, which allows our method to leverage unlabeled downstream task data. As shown in Fig. 3d, increasing the amount of unlabeled data for the priming stage (while keeping VR training fixed at 16-shot) steadily improves our method's performance. This ability to benefit from extra unlabeled data is a unique advantage of our approach compared to BMR baselines (BAR and BlackVIP), which cannot utilize such data in this manner.

## 6 CONCLUSION

In this work, we addressed the significant challenges inherent in adapting black-box Model-as-a-Service APIs: the prohibitive operational costs and optimization instability of the traditional ZOO-based methods, and their emerging ineffectiveness on modern, robust real-world black-box models. We introduced **PoRL** (Prime Once, then Reprogram Locally), a novel and highly efficient two-stage framework. By performing a one-time priming step to prepare a local encoder, followed by efficient visual reprogramming on the local model, PoRL achieves substantial improvements in both performance and real-world practicality. Our extensive experiments compellingly demonstrate that PoRL succeeds where prior methods fail, delivering significant accuracy gains on modern APIs like GPT-4o. Across ten standard benchmarks, it also consistently outperforms SOTA methods on both VMs and VLMs and dramatically reduces API calls by over 99%, while enabling entirely cost-free inference and subsequent offline deployment. These comprehensive findings underscore PoRL as a robust, economical, and highly effective alternative for black-box model adaptation, facilitating practical transfer learning in API-centric ecosystems and demonstrating that reprogrammability can be efficiently unlocked in a local model through a minimal, one-time priming interaction.

## ETHICS STATEMENT

The authors have adhered to the ICLR Code of Ethics. Our research is based on publicly available models and standard academic datasets, which do not contain personally identifiable information or sensitive content. We do not foresee any direct negative societal impacts or ethical concerns arising from our proposed method.

## REPRODUCIBILITY STATEMENT

To ensure full reproducibility, we will release our source code and experimental configurations publicly upon publication. Our experimental setup, including all models, datasets, and hyperparameter settings, is detailed in Section 5 and Appendix B. For our theoretical claims, a complete analysis with proofs is provided in Appendix C.

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

# Prime Once, then Reprogram Locally
## `Supplementary Materials`

This appendix provides comprehensive details supporting the main paper. Section A elaborates on model reprogramming techniques. Section B presents detailed experimental configurations, including dataset statistics and implementation details for all baselines and our PoRL method (Section B.3). Section C provides the complete theoretical analysis establishing performance bounds between service and local models. Additional experimental results are presented in Section D, including few-shot VM experiments (Section D.1), local model enhancement analysis (Section D.2), and detailed analysis of challenging VLM scenarios (Section D.7). Finally, Section F discusses the limitations of our approach, particularly in extreme data scarcity scenarios.

## A  DETAILS OF MODEL REPROGRAMMING

Model Reprogramming (MR) (Chen, 2024; Jia et al., 2022; Bahng et al., 2022; Zhang et al., 2022) is a technique that adapts pre-trained models, such as a source model $\mathcal{F}_S$, to new tasks without altering their underlying parameters. This is particularly useful when dealing with powerful models accessed as a service (MaaS) where direct modification is not possible. The core idea involves an input transformation function $g_{\text{in}}$ that adapts inputs $x^T$ from the downstream task domain $\mathcal{X}^T$ to the source domain $\mathcal{X}^S$ through a learnable prompt $\mathbf{P}$. Additionally, an output or label mapping function $g_{\text{out}}$ is employed to bridge the source label space $\mathcal{Y}^S$ and the target label space $\mathcal{Y}^T$. This approach aims to leverage the rich features learned by large pre-trained models for new, potentially resource-scarce tasks, by essentially "tricking" the model into performing a different function through these carefully crafted input and output manipulations.

### A.1  INPUT TRANSFORMATION

Input transformation (Tsao et al., 2024; Zhang et al., 2022; Oh et al., 2023; Tsai et al., 2020) in Model Reprogramming, denoted as $g_{\text{in}}(x^T|\mathbf{P})$, serves to bridge the domain gap between the downstream task and the source task the original model $\mathcal{F}_S$ was trained on. It introduces a learnable prompt or program $\mathbf{P}$ that modifies the downstream task inputs $x^T$ before they are fed to $\mathcal{F}_S$, aiming to make the downstream data compatible with its input expectations. Common approaches in Visual Reprogramming (VR) include padding-based VR, which adds trainable noise patterns to the outer frames of an image while preserving its integrity, and watermarking-based VR, where trainable noise patterns are overlaid directly onto the input images. In the context of BAR (Tsai et al., 2020), the input transformation takes the form of an "adversarial program" $P = \tanh(W \odot M)$, where $W$ represents learnable parameters and $M$ is a binary mask ensuring that the original embedded target data remains unchanged. This adversarial program $P$ is universal to all target data samples and is learned to make the model $\mathcal{F}_S$ produce outputs that can be mapped to the desired target task labels. The learning of these parameters, especially in black-box settings, often relies on zeroth-order optimization techniques.

### A.2  OUTPUT MAPPING

Output mapping (Chen et al., 2023; Cai et al., 2024a; Tsai et al., 2020; Kloberdanz et al., 2021), $g_{\text{out}}$, is a crucial step that translates the predictions from the pre-trained model's original label space $\mathcal{Y}^S$ to the downstream task's label space $\mathcal{Y}^T$, which are often distinct. This process is frequently designed to be gradient-free, meaning it does not introduce additional trainable parameters requiring backpropagation, thus preserving efficiency, especially for tasks with large label spaces. Existing gradient-free methods like Random Label Mapping (RLM) (Elsayed et al., 2018), Frequent Label Mapping (FLM) (Tsai et al., 2020), and Iterative Label Mapping (ILM) (Chen et al., 2023) typically establish a one-to-one correspondence. However, such one-to-one mappings can be limiting as they may overlook more complex, potentially many-to-many relationships between pre-trained and downstream labels. To address this, probabilistic and multi-label mapping strategies have been developed. Bayesian-guided Label Mapping (BLM), for example, constructs a probabilistic matrix quantifying pairwise relationships between pre-trained and downstream labels using Bayesian

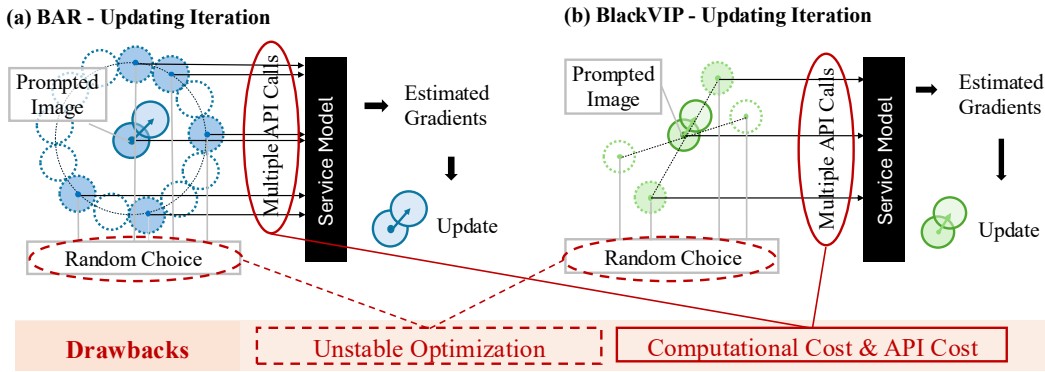

Figure 4: Illustration of Zeroth-Order Optimization (ZOO) techniques commonly used in black-box model reprogramming. **(a)** BAR with Randomized Gradient-Free (RGF) method estimates gradients by querying the model with random directional perturbations. **(b)** BlackVIP with Simultaneous Perturbation Stochastic Approximation with Gradient Correction (SPSA-GC) approximates gradients using only two model queries with a randomly generated perturbation vector. Both approaches suffer from high query complexity and noisy gradient estimates, leading to unstable and computationally intensive optimization, especially for high-dimensional prompts.

conditional probability, allowing for a flexible many-to-many mapping. Similarly, BAR (Tsai et al., 2020) can utilize multi-label mapping where multiple source labels map to a single target label, often determined by a frequency-based scheme from initial predictions.

The nature of $g_{\mathrm{out}}$ can differ for VMs and VLMs. For VMs, if label spaces differ, an explicit $g_{\mathrm{out}}$ like BLM is necessary, and in the PoRL framework, priming for VMs occurs in the service model's label probability space. For VLMs, $g_{\mathrm{out}}$ might be an identity mapping if label spaces align, or the VLM's text encoder can naturally transform outputs to the downstream task label space.

### A.3 TRAINING

The training process in model reprogramming focuses on learning the parameters of the input transformation $g_{\mathrm{in}}(\cdot|P)$, such as the adversarial program $P$, while the pre-trained source model $\mathcal{F}_S$ remains frozen. The objective is to minimize a loss function on the downstream task data using the transformed inputs and mapped outputs. In a white-box setting, where gradients from $\mathcal{F}_S$ are accessible, standard gradient-based optimization can efficiently update the transformation parameters. Our PoRL method leverages this by first priming a local, fully accessible model $\mathcal{F}_L$, then performing efficient white-box prompt learning. Conversely, when $\mathcal{F}_S$ is a black-box, Zeroth-Order Optimization (ZOO) techniques are employed. These methods estimate gradients using only the model's input-output responses, for instance, through random gradient-free (RGF) optimization (Tu et al., 2019; Liu et al., 2018), Simultaneous Perturbation Stochastic Approximation with Gradient Correction (SPSA-GC) (Spall, 1992; 1997), as shown in Fig. 4 and Fig. 5, black-box training presents significant difficulties: optimization can be unstable due to noisy gradient estimates; it incurs high computational costs and numerous API calls; it requires continuous API dependency for both training and inference; and the performance gains are often uncertain despite the substantial investment. Our PoRL approach aims to circumvent these challenges by shifting the reprogramming to a locally primed model.

## B EXPERIMENTAL SETTING

### B.1 DATASETS

**Datasets.** We evaluate our method on ten diverse datasets commonly used in transfer learning literature. These datasets span various domains including fine-grained categorization (Flowers102 (Nilsback & Zisserman, 2008), StanfordCars (Krause et al., 2013)), texture recognition (DTD (Cimpoi et al., 2014)), action recognition (UCF101 (Soomro et al., 2012)), food classification (Food101 (Bossard et al., 2014)), traffic sign recognition (GTSRB (Houben et al., 2013)), satellite

imagery (EuroSAT (Helber et al., 2019)), animal recognition (OxfordPets (Parkhi et al., 2012)), scene classification (SUN397 (Xiao et al., 2010)), and digit recognition (SVHN (Netzer et al., 2011)). Following established protocols in prior work (Oh et al., 2023), we adopt a few-shot learning setup with 16 randomly selected training examples per class, using the entire test set for evaluation when the service model is VLM. And full-shot learning is applied when the service model is VM, consistent with (Tsai et al., 2020). Table 7 provides detailed statistics for each dataset.

Table 7: Detailed dataset information.

| Dataset | Full-shot Training | 16-shot Training | Testing Set Size | Number of Classes |
|---|---|---|---|---|
| Flowers102 | 4,093 | 1,632 | 2,463 | 102 |
| DTD | 2,820 | 752 | 1,692 | 47 |
| UCF101 | 7,639 | 1,616 | 3,783 | 101 |
| Food101 | 50,500 | 1,616 | 30,300 | 101 |
| GTSRB | 39,209 | 688 | 12,630 | 43 |
| EuroSAT | 13,500 | 160 | 8,100 | 10 |
| OxfordPets | 2,944 | 592 | 3,669 | 37 |
| StanfordCars | 6,509 | 3,136 | 8,041 | 196 |
| SUN397 | 15,888 | 6,352 | 19,850 | 397 |
| SVHN | 73,257 | 160 | 26,032 | 10 |

## B.2 EXPERIMENTAL SCOPE AND RATIONALE

Our focus on image classification is a deliberate choice made to ensure a direct and rigorous comparison with the established art in Black-Box Model Reprogramming (BMR) (Tsai et al., 2020; Oh et al., 2023). The primary methods in this field, including our main baselines (BAR (Tsai et al., 2020), BlackVIP (Oh et al., 2023), LLM-Opt (Liu et al., 2024)) and other related works (Wang et al., 2024; Ouali et al., 2023; Guo et al., 2023; Kuna. et al., 2024; Yu et al., 2023) shown in Table 1, are all benchmarked on classification tasks for both standard Vision Models (VMs) and Vision-Language Models (VLMs). Adhering to this established protocol allows for a fair and robust evaluation of our framework, which is designed to provide an efficient reprogramming solution for both model types. Furthermore, classification remains a challenging and relevant testbed; recent work has shown that powerful Multimodal Large Language Models (MLLMs) e.g., LLaVA (Liu et al., 2023) often suffer from catastrophic forgetting, failing to retain the full classification performance of their underlying visual towers (Zhai et al.). This highlights the non-trivial nature of adapting these models for pure classification and validates the importance of our approach.

While extending PoRL to generative tasks like Visual Question Answering (VQA) is a valuable direction for future research, such tasks are outside the scope of this initial investigation. A core goal of our work is to support standard vision classifiers like ViT and ResNet, for which open-ended generative tasks are not directly applicable. By demonstrating PoRL's effectiveness on the fundamental task of classification—a common ground for both VMs and VLMs—we build a strong and reliable foundation for future extensions into more complex, multimodal reasoning tasks.

## B.3 BASELINES AND IMPLEMENTATION DETAILS

Our experiments compare PoRL against three primary black-box visual reprogramming baselines: BAR, BlackVIP, and LLM-Opt.

**BAR** (Tsai et al., 2020)[1] repurposes black-box models by learning a universal adversarial program that is added to target inputs. It relies on Randomized Gradient-Free (RGF) optimization, based on input-output responses, and employs a multi-label mapping scheme. For our VMs experiments, we utilize BLM[2] for BAR to ensure a fair comparison with other methods, including our own, which also uses BLM. We implemented BAR by referencing its official codebase and BlackVIP's reported

---

[1]https://github.com/yunyuntsai/Black-box-Adversarial-Reprogramming
[2]https://github.com/tmlr-group/BayesianLM

BAR implementation, adhering to the hyperparameters detailed in BlackVIP's appendix[3] for stable convergence. BAR uses focal loss as its learning objective.

**BlackVIP** (Oh et al., 2023)[4] enhances black-box visual prompting by introducing a Coordinator network to generate input-dependent visual prompts and employs Simultaneous Perturbation Stochastic Approximation with Gradient Correction (SPSA-GC) for optimization. We use the official BlackVIP codebase and strictly adhere to the hyperparameter settings reported in their paper and detailed in their configuration files, which cover learning rates and SPSA-GC specific parameters. BlackVIP utilizes cross-entropy loss. For adapting VLMs, all methods, including BlackVIP and BAR, are evaluated in a 16-shot learning setting (16 training samples per class), consistent with the setup in the BlackVIP paper.

**LLM-Opt** (Liu et al., 2024)[5] utilizes a large language model (e.g., GPT-4) as a black-box optimizer to find effective text prompts for a target VLM. The method employs an automated "hill-climbing" procedure, where it provides the LLM optimizer with both high- and low-performing prompts as textual feedback to guide the search for better candidates (Liu et al., 2024). However, this approach has two significant limitations. First, it introduces a second, costly API dependency for the LLM optimizer, which can amount to over $100 in API fees for a single experimental run (Liu et al., 2024). Second, because the method exclusively optimizes the text prompt, its applicability is restricted to VLMs and it cannot be used to adapt standard VMs, which are a key focus of our work.

For our proposed **PoRL** method, the visual reprogramming (VR) components are implemented leveraging the publicly available codebase of BLM. The priming stage, transferring from the service API to the local model's classification head, is performed for 100 epochs. The subsequent white-box VR on the primed local model is conducted for 200 epochs. Both stages use the Adam optimizer with learning rates of 0.001 for KD and 0.01 for local VR, respectively. The local VR part of PoRL employs cross-entropy loss. PoRL follows the 16-shot protocol for VLM experiments and operates in a full-shot setting for VM experiments. Following (Oh et al., 2023), pre-trained Vision Transformer (ViT) models and encoders used in our experiments are sourced from `timm` (Wightman, 2019), ResNet models are from `PyTorch` (Paszke et al., 2019), and CLIP models are from the official `CLIP` repository (Radford et al., 2021).

### B.4 Clarification of VLM and VM Evaluation Setups

Our experiments distinguish between two primary evaluation setups: one for Vision-Language Models (VLMs) and another for standard Vision Models (VMs). The core differences lie in how each model type handles the output space of the downstream task and the data settings used for evaluation, which were chosen to ensure fair and direct comparisons with established baselines.

**VLM Evaluation Setup.** In our experiments, the VLM setup primarily involves models like CLIP, which is consistent with the standard set by our main baseline, BlackVIP. The key characteristics are:

- **Output Handling:** VLMs utilize a text encoder to perform zero-shot classification without a fixed output head. This means their output space is naturally aligned with the downstream task's text labels. Consequently, a separate or complex label mapping mechanism is not required, simplifying the adaptation process.

- **Data Setting:** To ensure a direct comparison, the evaluation follows BlackVIP's established protocol, which uses a **16-shot** learning setting. The black-box model reprogramming for all compared methods is learned using only these few-shot labeled samples.

**VM Evaluation Setup.** The VM setup uses standard, pre-trained vision models with a fixed output vocabulary, such as an ImageNet-pretrained ViT-B/16 or ResNet101. This setup presents a more significant challenge:

- **The Challenge of Label Space Mismatch:** Unlike VLMs, these models face a fundamental label space mismatch. Their fixed output vocabulary (e.g., 1000 ImageNet classes) does

---

[3]https://github.com/changdaeoh/BlackVIP/blob/main/docs/configuration.md
[4]https://github.com/changdaeoh/BlackVIP
[5]https://llm-can-optimize-vlm.github.io

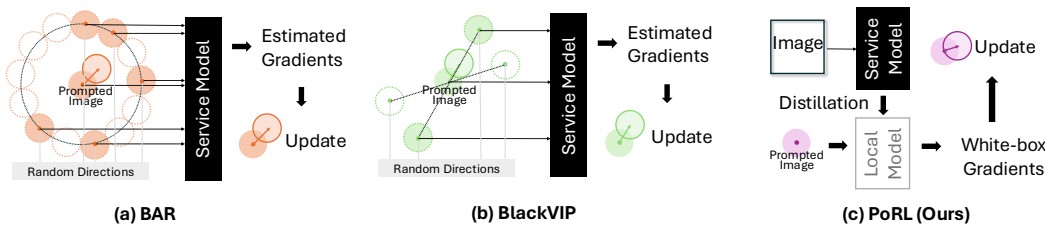

Figure 5: Comparison of black-box visual reprogramming approaches. **(a)** BAR and **(b)** BlackVIP rely on Zeroth-Order Optimization (ZOO) by repeatedly querying the service model with perturbed inputs (e.g., using random directions) to estimate gradients for updating the visual prompt. These methods suffer from high API call costs and potentially unstable optimization. **(c)** Our PoRL method performs a one-time priming from the service model to a local model. Subsequent visual prompt optimization occurs efficiently on this local model using white-box gradients, eliminating further API calls and enabling stable, cost-effective adaptation.

  not align with the labels of the downstream task (e.g., Flowers102). This is a "non-trivial problem" that prior work like BlackVIP explicitly avoided.

- **PoRL's Solution:** Our framework directly solves this challenge by incorporating Bayesian-guided Label Mapping (BLM) to bridge the source and target label spaces. In this setup, PoRL's priming stage occurs within the service model's source label space, as it has no inherent knowledge of the target class names.

- **Data Setting:** Following the standard for BMR on vision models established by BAR, these experiments are conducted in a **full-shot** setting. For completeness, a few-shot VM setting was also explored in our appendix.

## C  DETAILS OF THEORETICAL ANALYSIS

In this section, we provide a theoretical analysis to understand the effectiveness of our proposed method, PoRL. Our framework involves two primary components affecting the final performance on the downstream task: priming from the service model $\mathcal{F}_S$ to the local encoder $\mathcal{F}_L$, and visual reprogramming (VR) applied to the local model using prompt $\mathbf{P}$. We begin our analysis by considering the scenario where the model's output logits are directly aligned with the target labels, effectively assuming the label mapping $g_{\text{out}}$ is an identity function due to the text encoder.

**Notations:** Let $\mathcal{D}^T$ represent the downstream task data distribution $p(x, y)$ over inputs $x \in \mathcal{X}^T$ and labels $y \in \mathcal{Y}^T$. Let $\mathcal{F}_S : \mathcal{X}^S \to \mathbb{R}^{K^S}$ be the black-box service model and $\mathcal{F}_L : \mathcal{X}^S \to \mathcal{Z}$ be the local surrogate model. After priming, $\mathcal{F}_L$ includes the learned linear layer mapping inputs to the downstream task's logit space $\mathcal{Z} = \mathbb{R}^{K^T}$. The visual reprogramming involves an input transformation $g_{\text{in}}(x, p)$ parameterized by $p$. $\ell : \mathcal{Z} \times \mathcal{Y}^T \to \mathbb{R}_{\geq 0}$ is cross-entropy loss.

Let $p_S(x)$ and $p_L(x)$ denote the probability distributions generated by the service model $\mathcal{F}_S$ and the local model $\mathcal{F}_L$ for an input $x$, respectively. For the purpose of this theoretical analysis, we introduce logits. Let $z_S(x)$ and $z_L(x)$ be the logits produced by the service model $\mathcal{F}_S$ and the local model $\mathcal{F}_L$ for input $x$ *before* visual reprogramming. It is important to note that this assumption of direct logit access from $\mathcal{F}_S$ is made for theoretical tractability and differs from the practical setting in the main paper, where only probabilities are assumed to be accessible from the service model. Let $z_S^*(x, \mathbf{Q}^*) = \mathcal{F}_S(g_{\text{in}}(x, \mathbf{Q}^*))$ and $z_L^*(x, \mathbf{P}^*) = \mathcal{F}_L(g_{\text{in}}(x, \mathbf{P}^*))$ be the logits produced by the service model and local model, respectively, for input $x$ *after* visual reprogramming with their respective optimal prompts $\mathbf{Q}^*$ and $\mathbf{P}^*$.

The downstream risk for the local model $\mathcal{F}_L$ *without* VR is:

$$\mathcal{R}_L(\mathcal{D}^T) := \mathbb{E}_{(x,y) \sim \mathcal{D}^T}[\ell(z_L(x), y)]$$

The downstream risk for the local model $\mathcal{F}_L$ *after* applying VR with its optimal prompt $\mathbf{P}^*$ is:

$$\mathcal{R}_L(\mathcal{D}^T, \mathbf{P}^*) := \mathbb{E}_{(x,y) \sim \mathcal{D}^T}[\ell(z_L^*(x, \mathbf{P}^*), y)]$$

Similarly, for the service model $\mathcal{F}_S$, the downstream risk *without* VR is:

$$\mathcal{R}_S(\mathcal{D}^T) := \mathbb{E}_{(x,y)\sim\mathcal{D}^T}[\ell(z_S(x), y)].$$

And the downstream risk for the service model $\mathcal{F}_S$ *after* applying VR with its optimal prompt $\mathbf{Q}^*$ is:

$$\mathcal{R}_S(\mathcal{D}^T, \mathbf{Q}^*) := \mathbb{E}_{(x,y)\sim\mathcal{D}^T}[\ell(z_S^*(x, \mathbf{Q}^*), y)].$$

**Definitions and Assumptions.** We introduce the following definitions and assumptions for our theoretical analysis.

**Definition 1.** ($\epsilon$-Faithful Priming). The priming from $\mathcal{F}_S$ to $\mathcal{F}_L$ is considered $\epsilon$-faithful if the expected $L_1$ norm of the difference between their output logits is bounded by $\epsilon \geq 0$, both before and after applying their respective optimal visual reprogramming prompts:

$$\mathbb{E}_{(x,y)\sim\mathcal{D}^T}[\|z_S(x) - z_L(x)\|_1] \leq \epsilon$$

and

$$\mathbb{E}_{(x,y)\sim\mathcal{D}^T}[\|z_S^*(x, \mathbf{Q}^*) - z_L^*(x, \mathbf{P}^*)\|_1] \leq \epsilon.$$

This implies that the priming is sufficiently effective such that the logit distributions of the service and local models are closely aligned. A smaller $\epsilon$ indicates a more faithful priming.

**Assumption 1.** *(Service Model Superiority). The service model $\mathcal{F}_S$ is inherently more powerful or better aligned with the downstream task distribution $\mathcal{D}^T$ compared to the local model $\mathcal{F}_L$, both before and after optimal visual reprogramming. Formally:*

$$\mathcal{R}_S(\mathcal{D}^T) \leq \mathcal{R}_L(\mathcal{D}^T)$$

*and*

$$\mathcal{R}_S(\mathcal{D}^T, \mathbf{Q}^*) \leq \mathcal{R}_L(\mathcal{D}^T, \mathbf{P}^*).$$

*This is a natural assumption that motivates the use of the service model.*

**Assumption 2.** *($\epsilon$-Faithful Priming). The priming process is effective, resulting in the local model $\mathcal{F}_L$ closely mimicking the logit distributions of the service model $\mathcal{F}_S$, both immediately after priming (before VR) and after both models have been optimally reprogrammed for the downstream task.*

### C.1 SERVICE AND LOCAL MODEL PERFORMANCE DIFFERENCE BOUND

This section establishes the theoretical foundation for understanding the performance relationship between service and local models in our PoRL framework. We begin by proving that the cross-entropy loss function exhibits Lipschitz continuity with respect to logit differences (Lemma 1), which provides the mathematical basis for our subsequent analysis. Building on this property, we derive performance difference bounds (Lemma 2) that quantify how faithful priming affects the performance gap between models. Finally, we establish tight bounds on service model performance relative to the primed local model (Theorem 3), demonstrating that faithful priming enables the local model to closely approximate the service model's capabilities both before and after optimal visual reprogramming.

**Lemma 1.** *(Lipschitz Continuity of Cross-Entropy Loss with respect to Logits). The cross-entropy loss function $\ell(z, y) = -\sum_{j=1}^{K^T} y_j \log(p_j(z))$, where $p_j(z)$ are softmax probabilities derived from logits $z \in \mathbb{R}^{K^T}$ and $y$ is a one-hot true label vector, is Lipschitz continuous with respect to the logits $z$. The specific Lipschitz constant depends on the norm used to measure the difference between logit vectors. Specifically: $\ell(z, y)$ is 1-Lipschitz with respect to the $L_1$ norm of the logits:*

$$|\ell(z_1, y) - \ell(z_2, y)| \leq \|z_1 - z_2\|_1,$$

*for any two logit vectors $z_1, z_2$. These constants do not explicitly depend on the number of classes $K^T$ (for $K^T \geq 2$).*

*Proof.* The gradient of the cross-entropy loss $\ell(z, y)$ with respect to the logits $z$ is given by:

$$\nabla_z \ell(z, y) = p(z) - y,$$

where $p(z)$ is the vector of softmax probabilities derived from $z$, and $y$ is the one-hot true label vector. A differentiable function $f(x)$ is $L$-Lipschitz with respect to a norm $\|\cdot\|_p$ if the dual norm $\|\cdot\|_q$ of

its gradient is bounded by $L$ (i.e., $\|\nabla f(x)\|_q \leq L$), by Hölder's inequality. For the $L_1$ norm of logits ($p = 1$, dual norm $q = \infty$): We examine the $L_\infty$ norm of the gradient:

$$\|\nabla_z \ell(z, y)\|_\infty = \|p(z) - y\|_\infty = \max_{j=1,\ldots,K^T} |p_j(z) - y_j|.$$

Let $k$ be the index of the true class ($y_k = 1$, and $y_j = 0$ for $j \neq k$). Then $|p_k(z) - 1| = 1 - p_k(z) \leq 1$. For $j \neq k$, $|p_j(z) - 0| = p_j(z) \leq 1$. Thus, $\|\nabla_z \ell(z, y)\|_\infty \leq 1$. This establishes $L = 1$.

$\square$

**Lemma 2.** *(Performance Difference Bound due to Priming Faithfulness). Under Assumption 2 ($\epsilon$-Faithful Priming), the difference in performance between the local model $\mathcal{F}_L$ and the service model $\mathcal{F}_S$ is bounded by $\epsilon$ as follows:*

*1. Before visual reprogramming:*

$$\mathcal{R}_L(\mathcal{D}^T) - \mathcal{R}_S(\mathcal{D}^T) \leq \epsilon;$$

*2. After optimal visual reprogramming with prompts $\mathbf{P}^*$ for $\mathcal{F}_L$ and $\mathbf{Q}^*$ for $\mathcal{F}_S$:*

$$\mathcal{R}_L(\mathcal{D}^T, \mathbf{P}^*) - \mathcal{R}_S(\mathcal{D}^T, \mathbf{Q}^*) \leq \epsilon.$$

*Proof.* We will prove the first part of the lemma regarding performance before visual reprogramming. The proof for the second part (after optimal visual reprogramming) follows an identical structure, applying the arguments to $z_L^*(x, \mathbf{P}^*)$ and $z_S^*(x, \mathbf{Q}^*)$ and using the corresponding condition from Assumption 2.

The difference in downstream risks between the local model $\mathcal{F}_L$ and the service model $\mathcal{F}_S$ is:

$$\mathcal{R}_L(\mathcal{D}^T) - \mathcal{R}_S(\mathcal{D}^T) = \mathbb{E}_{(x,y)\sim\mathcal{D}^T}[\ell(z_L(x), y)] - \mathbb{E}_{(x,y)\sim\mathcal{D}^T}[\ell(z_S(x), y)]$$
$$= \mathbb{E}_{(x,y)\sim\mathcal{D}^T}[\ell(z_L(x), y) - \ell(z_S(x), y)].$$

By Lemma 1, the cross-entropy loss $\ell(z, y)$ is 1-Lipschitz continuous with respect to the $L_1$ norm of its logit inputs. Therefore, for any specific sample $(x, y)$:

$$\ell(z_L(x), y) - \ell(z_S(x), y) \leq |\ell(z_L(x), y) - \ell(z_S(x), y)| \leq \|z_L(x) - z_S(x)\|_1.$$

Taking the expectation over the data distribution $\mathcal{D}^T$:

$$\mathbb{E}_{(x,y)\sim\mathcal{D}^T}[\ell(z_L(x), y) - \ell(z_S(x), y)] \leq \mathbb{E}_{(x,y)\sim\mathcal{D}^T}[\|z_L(x) - z_S(x)\|_1].$$

From Assumption 2 ($\epsilon$-Faithful Priming), we have the condition that $\mathbb{E}_{(x,y)\sim\mathcal{D}^T}[\|z_S(x) - z_L(x)\|_1] \leq \epsilon$. Substituting this into the inequality:

$$\mathcal{R}_L(\mathcal{D}^T) - \mathcal{R}_S(\mathcal{D}^T) \leq \epsilon.$$

$\square$

**Theorem 3.** *(Bounding Service Model Performance). Combining Lemma 2 with Assumption 1 (Service Model Superiority), we can establish bounds on the performance of the service model $\mathcal{F}_S$ relative to the primed local model $\mathcal{F}_L$:*

*1. Before visual reprogramming:*

$$\mathcal{R}_L(\mathcal{D}^T) - \epsilon \leq \mathcal{R}_S(\mathcal{D}^T) \leq \mathcal{R}_L(\mathcal{D}^T); \tag{4}$$

*2. After optimal visual reprogramming:*

$$\mathcal{R}_L(\mathcal{D}^T, \mathbf{P}^*) - \epsilon \leq \mathcal{R}_S(\mathcal{D}^T, \mathbf{Q}^*) \leq \mathcal{R}_L(\mathcal{D}^T, \mathbf{P}^*). \tag{5}$$

*Proof.* By Lemma 2, we have $\mathcal{R}_L(\mathcal{D}^T) - \mathcal{R}_S(\mathcal{D}^T) \leq \epsilon$, which rearranges to $\mathcal{R}_L(\mathcal{D}^T) - \epsilon \leq \mathcal{R}_S(\mathcal{D}^T)$. From Assumption 1, we have $\mathcal{R}_S(\mathcal{D}^T) \leq \mathcal{R}_L(\mathcal{D}^T)$. Combining these two inequalities gives:

$$\mathcal{R}_L(\mathcal{D}^T) - \epsilon \leq \mathcal{R}_S(\mathcal{D}^T) \leq \mathcal{R}_L(\mathcal{D}^T).$$

The reprogrammed case can be shown similarly. $\square$

Table 8: Accuracy and Efficiency comparison using RN101 (Service) in Full-shot setting.

| Method | Flowers | DTD | UCF | Food | GTSRB | EuroSAT | Pets | Cars | SUN | SVHN | Avg. | #API (M) | Time (h) |
|---|---|---|---|---|---|---|---|---|---|---|---|---|---|
| VR (white-box) | 42.3 | 44.4 | 34.4 | 25.4 | 53.1 | 85.0 | 73.9 | 5.0 | 20.5 | 75.1 | 45.9 | 43.4 | 21.9 |
| BAR | 16.9 | 25.3 | 30.2 | 13.9 | 22.5 | 45.7 | 28.7 | 1.5 | 10.7 | 42.1 | 23.8 | 1,724.2 | 313.5 |
| BlackVIP (RN50) | 21.7 | 39.6 | 31.4 | 14.9 | 26.8 | 68.3 | 61.4 | 3.5 | 12.3 | 42.6 | 32.3 | 2,586.2 | 320.5 |
| **PoRL (RN50)** | **41.3** | **42.3** | **37.2** | **25.3** | **48.4** | **84.3** | **73.7** | **5.1** | **19.7** | **72.9** | **45.0** | **0.2** | **8.7** |
| BlackVIP (ViT-B/32) | 26.7 | 37.7 | 30.2 | 13.3 | 33.8 | 67.3 | 57.9 | 3.0 | 14.5 | 34.5 | 31.9 | 2,586.2 | 418.0 |
| **PoRL (ViT-B/32)** | **51.3** | **45.8** | **41.5** | **30.2** | **57.9** | **94.9** | **63.3** | **5.1** | **25.6** | **83.6** | **49.9** | **0.2** | **22.3** |

Table 9: Accuracy comparison using ViT-B/16 (Service) and ViT-B/32 (Local) in 16-shot setting.

| Method | Flowers | DTD | UCF | Food | GTSRB | EuroSAT | Pets | Cars | SUN | SVHN | Avg. |
|---|---|---|---|---|---|---|---|---|---|---|---|
| VR (white-box) | 55.0 | 44.7 | 42.0 | 18.0 | 17.4 | 70.6 | 70.7 | 5.7 | 29.6 | 28.4 | 38.2 |
| BAR | 9.7 | 16.3 | 23.5 | 8.1 | 4.8 | 34.2 | 21.5 | 1.0 | 7.6 | 19.6 | 14.6 |
| BlackVIP | 16.1 | **36.1** | 33.8 | 12.5 | 9.1 | 54.9 | 54.5 | 3.9 | 22.1 | 16.2 | 25.9 |
| **PoRL (Ours)** | **46.0** | 35.5 | **34.3** | **12.7** | **19.6** | **71.1** | 54.3 | **4.2** | **23.0** | **27.3** | **32.8** |

The above theoretical analysis provides solid justification for PoRL's effectiveness by showing that faithful knowledge priming ($\epsilon$-faithful) directly translates to bounded performance differences between service and local models. Theorem 3's key insight reveals that traditional black-box methods expend significant computational resources attempting to optimize the service model's performance directly, while PoRL transforms this challenge by first achieving faithful priming (small $\epsilon$) and then efficiently optimizing the local model using stable first-order methods. This theoretical framework validates our empirical findings and explains why PoRL can achieve competitive performance with dramatically reduced API calls and computational costs.

# D   ADDITIONAL EXPERIMENTAL RESULTS

## D.1   VM AS A SERVICE MODEL IN FEW-SHOT SETTING

Table 9 details the performance of VM adaptation using ViT-B/16 as the service model and ViT-B/32 as the local model within a challenging 16-shot per class setting. This contrasts with the full-shot experiments for VMs presented in the main paper (Tables 8 and 3), which are generally preferred for VMs due to their more limited generalization compared to VLMs like CLIP. These few-shot results, however, provide a valuable benchmark for performance in data-scarce conditions. As anticipated, all evaluated methods show a decline in performance relative to the full-shot scenario. Notably, the black-box reprogramming methods BAR and BlackVIP achieve average accuracies of only 14.6% and 25.9%, respectively. Despite this challenging low-data regime, our PoRL method demonstrates superior performance, achieving an average accuracy of 32.8%. This represents a significant average improvement of +18.2 percentage points over BAR and +6.9 percentage points over BlackVIP. This consistent outperformance, even with extremely limited training data, underscores PoRL's effectiveness in robustly transferring knowledge from black-box service models and significantly enhancing local model reprogramming capabilities across diverse datasets and varying data availability.

## D.2   PoRL'S IMPACT ON LOCAL MODEL PERFORMANCE

To quantify PoRL's enhancement of local model capabilities, we compare it against directly performing white-box VR on the local model (termed "Local VR"). For PoRL, we assume a pre-trained local encoder, consistent with methods like BlackVIP. For this evaluation of the local encoder only, we additionally assume a pre-trained linear layer on the source domain. As shown in Table 10 for VLMs (CLIP ViT-B/16 service, ViT-B/16 local, 16-shot), PoRL achieves a 65.4% average accuracy, a substantial $+27.2\%$ improvement over Local VR's 38.2%. This highlights that PoRL's priming significantly boosts the local ViT-B/16's reprogrammability beyond its standalone capacity.

A similar advantage for PoRL is observed with VMs in the full-shot setting, as detailed in Table 11. For instance, with a ViT-B/16 service model, PoRL improves upon Local VR by $+5.1\%$ (50.4% vs. 45.3%) when using a ViT-B/32 local model, and by $+2.0\%$ (45.9% vs. 43.9%) with an RN50 local model. Consistent gains are also seen with an RN101 service model. These results across both VLM

Table 10: Accuracy comparison between Local VR (ViT-B/16) and our method on VLMs using CLIP ViT-B/16 (Service) and ViT-B/16 (Local) in a 16-shot setting.

| Method | Flowers | DTD | UCF | Food | GTSRB | EuroSAT | Pets | Cars | SUN | SVHN | Avg. |
|---|---|---|---|---|---|---|---|---|---|---|---|
| Zero-shot | 71.3 | 43.9 | 66.9 | **85.9** | 21.0 | 47.9 | **89.1** | **65.2** | 62.6 | 17.9 | 57.2 |
| Local VR | 55.0 | 44.7 | 42.0 | 18.0 | 17.4 | 70.6 | 70.7 | 5.7 | 29.6 | 28.4 | 38.2 |
| **PoRL (Ours)** | **86.6** | **48.2** | **67.1** | 68.8 | **39.4** | **85.7** | 88.9 | 43.2 | **62.8** | **63.2** | **65.4** |

Table 11: Accuracy comparison between Local VR and our method on VMs in Full-shot setting.

| Service | Local | Local VR | Ours |
|---|---|---|---|
| ViT-B/16 | ViT-B/32 | 45.3 | **50.4** |
|  | RN50 | 43.9 | **45.9** |
| RN101 | ViT-B/32 | 45.3 | **49.9** |
|  | RN50 | 43.9 | **45.0** |

and VM configurations underscore the critical role of PoRL's initial priming phase. By effectively transferring knowledge from the more powerful service model, PoRL significantly elevates the local model's baseline performance and its potential for reprogramming, demonstrating a more effective utilization of combined model strengths.

### D.3 DISSECTING THE SOURCE OF PERFORMANCE GAIN

The superior performance of PoRL stems not from a higher parameter count or the mere inclusion of local tuning, but from its novel two-stage framework. We demonstrate that the initial priming stage is an indispensable component and that the overall design is highly parameter-efficient.

First, an analysis of parameter efficiency reveals that for ZOO-based methods, a larger number of trainable parameters does not guarantee better performance and can even be detrimental. As shown in Table 12, a baseline using SPSA-GC with a large visual prompt (69K parameters) achieves a lower accuracy than more compact models. This is likely because noisy gradient estimates are less stable in higher-dimensional spaces. In contrast, PoRL's stable, white-box optimization allows for the effective tuning of a compact prompt, confirming its effectiveness is due to a superior optimization strategy, not parameter quantity.

Second, the performance gain is critically dependent on the initial priming stage. Our component analysis on EuroSAT (Table 6 in the main paper) shows that local visual reprogramming alone (Local VR) achieves 70.6% accuracy. The full PoRL method, which includes priming, elevates this to 85.7%, a substantial +15.1% improvement demonstrating that the knowledge transferred during priming is essential for unlocking the local model's full potential. To further validate this, we created an enhanced version of BlackVIP that incorporates a local pre-training stage. To mimic a local pre-training stage, we trained its prompt-generating Coordinator decoder with a reconstruction loss, as direct supervision is not possible without a ground-truth prompt. As shown in Table 13, this local optimization provides only a marginal improvement to BlackVIP (+0.8%). PoRL still significantly outperforms this enhanced baseline by +11.6%, confirming that the performance gain originates from our superior two-stage framework, where priming makes the local model significantly more amenable to reprogramming.

### D.4 PERFORMANCE SCALING WITH DATA AVAILABILITY

A key question is whether PoRL's advantage over a strong, locally-trained baseline persists as more training data becomes available. To investigate this, we conducted an ablation study on the EuroSAT dataset, varying the number of training samples from 4 to 32 shots. The results in Table 14 show that PoRL consistently outperforms both the ZOO-based baseline (BlackVIP) and a strong local baseline (Local VR+LP) across all data settings.

While the local baseline's performance improves steadily with more data, it never surpasses PoRL. Our method is not only competitive in low-data regimes but also scales more effectively as data increases. For instance, at 32 shots, PoRL achieves 91.6% accuracy, maintaining a significant +5.1%

Table 12: Comparison of trainable parameters and accuracy on the EuroSAT dataset. More parameters do not correlate with better performance for ZOO-based methods.

| Method | # Trainable Params | Accuracy (%) |
|---|---|---|
| VP w/ SPSA-GC | 69K | 70.9 |
| BAR | 37K | 77.3 |
| BlackVIP | 9K | 73.3 |
| **PoRL (Ours)** | **21K** | **85.7** |

Table 13: Comparison with an enhanced BlackVIP baseline on the EuroSAT dataset. The results confirm that PoRL's performance gain is primarily from its superior two-stage framework.

| Method | Accuracy (%) |
|---|---|
| Local VR | 70.6 |
| BlackVIP | 73.3 |
| BlackVIP w/ Local Tuning | 74.1 |
| **PoRL (Ours)** | **85.7** |

lead over the 86.5% from the local baseline. This confirms that the initial priming stage provides a durable advantage that local training alone cannot replicate, equipping the local model with a superior foundation that enables a higher performance ceiling.

### D.5 SYNERGISTIC EFFECT OF COMBINING MODEL KNOWLEDGE

In certain configurations, PoRL's performance can surpass that of a white-box reprogramming approach on the service model itself. This outcome stems from a synergistic effect, where our framework effectively combines the distinct knowledge of the powerful service model with the unique inductive biases of the local model. Unlike methods (Oh et al., 2023) that use the local model as a simple feature generator, PoRL's two-stage design first primes and then leverages the local model's capabilities during reprogramming. This process allows the final model to correctly classify samples that neither the service model nor the standalone local model could handle individually.

This synergistic effect is demonstrated quantitatively in our analysis of the Flowers102 dataset (Table 15), where we compare scenarios with service models of varying strength. When the service model (RN101) has capabilities comparable to the local model, PoRL's final accuracy surpasses both, largely by learning to classify an additional 4.8% of samples that were incorrect for both models initially. Conversely, when the service model (ViT-B/16) is significantly stronger, PoRL's role shifts to highly effective knowledge transfer, yielding a remarkable +14.9% absolute improvement over reprogramming the local model alone. This analysis confirms that PoRL facilitates a potent combination of knowledge, creating synergistic outcomes not possible with other approaches.

### D.6 PERFORMANCE ON LARGE-SCALE BENCHMARKS: IMAGENET

To further validate our method's effectiveness on a large-scale, stable benchmark, we conducted experiments on ImageNet in a 16-shot setting, using CLIP ViT-B/16 as the service model. The results, presented in Table 16, demonstrate PoRL's significant superiority. While prior ZOO-based methods like BAR and VP w/ SPSA-GC degrade performance compared to the zero-shot baseline, and the SOTA BlackVIP provides only a marginal +0.4% gain, PoRL achieves a remarkable 80.1% accuracy. This represents a substantial +13.4% improvement over the zero-shot baseline and a +13.0% gain over the next best competitor, BlackVIP. Notably, PoRL even surpasses the white-box reprogramming baseline, further highlighting the powerful synergistic effect of combining the service model's knowledge with the local model's inductive biases. This is particularly notable as it demonstrates that when a strong local model is available, PoRL can effectively employ this advantage to achieve performance superior to both the service model and other local-model-based methods. In contrast, the BlackVIP baseline, despite utilizing the same local encoder architecture, fails to leverage its capabilities, providing only a negligible improvement. These compelling results on a challenging, large-scale dataset confirm that our conceptual shift away from the ZOO-based paradigm is a more robust and effective strategy for black-box model adaptation.

Table 14: Accuracy (%) comparison on the EuroSAT dataset across different few-shot settings. PoRL consistently outperforms the strong local baseline (Local VR+LP) as data availability increases.

| Method | 4-shot | 8-shot | 16-shot | 32-shot |
|---|---|---|---|---|
| BlackVIP | 69.3 | 71.7 | 73.3 | 72.9 |
| Local VR+LP | 59.7 | 68.1 | 80.1 | 86.5 |
| **PoRL (Ours)** | **66.1** | **73.4** | **85.7** | **91.6** |

Table 15: Illustrative Breakdown of PoRL's Performance Composition on Flowers102. The local model for both scenarios is ViT-B/32 (White-box Acc: 38.7%). "Kept correct" refers to samples the local model classified correctly that PoRL also classifies correctly. "Newly learned" refers to samples the local model misclassified that PoRL now classifies correctly.

| Initial State | Service: RN101 (42.3%) | | Service: ViT-B/16 (69.6%) | |
|---|---|---|---|---|
| | % of Dataset | How PoRL Performed | % of Dataset | How PoRL Performed |
| Both Service & Local Correct | 30.0% | 29.5% kept correct | 35.0% | 34.0% kept correct |
| Only Local Correct | 8.7% | 7.0% kept correct | 3.7% | 0.5% kept correct |
| Only Service Correct | 12.3% | 10.0% newly learned | 34.6% | 19.0% newly learned |
| Both Incorrect | 49.0% | 4.8% newly learned | 26.7% | 0.1% newly learned |
| **PoRL Final Accuracy** | **51.3%** | | **53.6%** | |

### D.7 PERFORMANCE ANALYSIS ON CHALLENGING VLM SCENARIOS

While PoRL demonstrates considerable average performance improvements with unparalleled efficiency, its efficacy with VLMs like CLIP as a service can exhibit variability on particularly challenging datasets, notably Food101 and Cars, as indicated in Table 2. The inherent difficulty of these datasets for reprogramming is underscored by the performance of even highly capable baselines. For instance, white-box VR performed directly on the CLIP ViT-B/16 service model achieves 81.6% on Food101, which is below the zero-shot performance of 85.9%. On Cars, white-box VR obtains 66.2%, only a marginal improvement over the 65.2% zero-shot accuracy. This suggests a ceiling for reprogramming efficacy on these complex domains, even with full model access. Consequently, ZOO-based black-box methods like BAR and BlackVIP also struggle; on Food101, BAR (84.4%) and BlackVIP (85.9%) offer negligible to no improvement over zero-shot performance. Similarly, on Cars, BAR (63.0%) underperforms the zero-shot baseline, and BlackVIP (65.4%) provides only a minimal gain. These results highlight that substantial API call volumes and computational efforts by existing black-box methods do not reliably translate into significant performance gains on such challenging datasets.

In this context, PoRL's performance (68.8% on Food101 and 43.2% on Cars in Table 2) is logically influenced by the capabilities of the local ViT-B/16 model. Given that our approach entirely avoids API calls during inference and subsequent reprogramming, its success is intrinsically tied to how well the primed knowledge empowers the local model for the specific downstream task. On these particularly demanding datasets, the primed local model may not fully capture the intricate features that the larger service model leverages for its strong zero-shot performance, leading to a performance gap. However, it is crucial to evaluate PoRL's contribution in terms of enhancing the local model's standalone reprogrammability. As shown in Table 10, direct Local VR on the ViT-B/16 achieves only 18.0% on Food101 and a mere 5.7% on Cars. PoRL elevates these figures to 68.8% and 43.2% respectively, marking substantial improvements of +50.8% and +37.5%. This significant enhancement of the local model's utility, achieved with minimal, one-time API interaction and drastically reduced computation, is of immense practical value, particularly in scenarios where continuous and costly reliance on service APIs is untenable. Addressing the remaining performance gap on these specific challenging datasets, possibly through advancements in primed techniques or by employing more capable local architectures, presents an interesting direction for future research.

### D.8 RESULTS ON MLLMs AND REAL-WORLD APIs

To validate PoRL's practical effectiveness on modern models, we conducted a targeted evaluation using the EuroSAT 16-shot benchmark on three distinct services: the open-source MLLM LLaVA, the proprietary VLM GPT-4o, and the commercial VM API Clarifai. Since MLLMs do not produce

Table 16: Accuracy (%) comparison on ImageNet in a 16-shot setting with CLIP ViT-B/16 as the service model. PoRL significantly outperforms all baselines.

| Method | ImageNet Acc (%) | Gain over Zero-shot (%) |
|---|---|---|
| VR (white-box) | 67.4 | +0.7 |
| Zero-shot | 66.7 | - |
| BAR | 64.6 | -2.1 |
| VP w/ SPSA-GC | 62.3 | -4.4 |
| BlackVIP | 67.1 | +0.4 |
| **PoRL (Ours)** | **80.1** | **+13.4** |

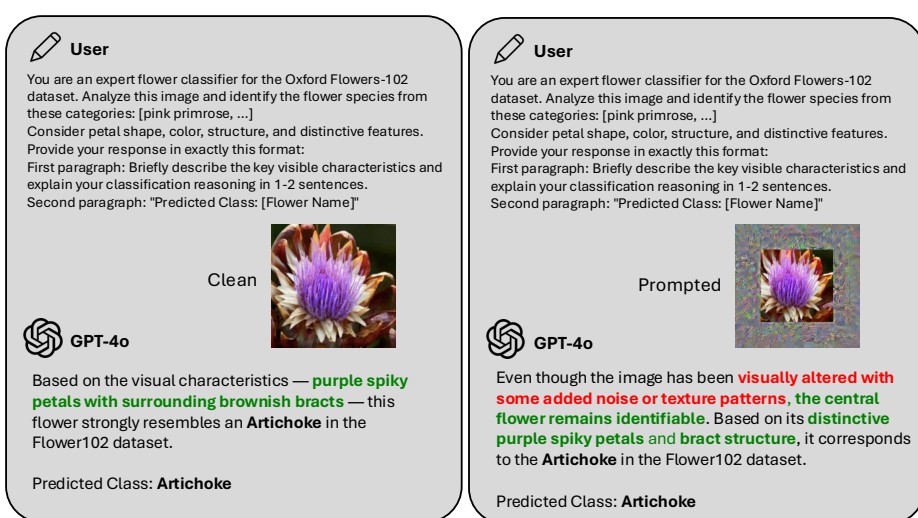

Figure 6: An example of GPT-4o's robustness to the input perturbations used in reprogramming. When presented with a clean image (left) and a prompted image (right), GPT-4o's textual reasoning explicitly acknowledges the visual alteration but correctly identifies the flower's key features in both cases. This demonstrates that the model is often invariant to the input noise that ZOO-based methods rely on, motivating a strategic shift in adaptation methods.

output probabilities by default, we instructed the models to act as a classifier and return a confidence score for each class (e.g., *"For the classes [...], provide a confidence score from 0 to 1 for each."*). Our evaluation reveals a critical limitation of traditional BMR on these services. As illustrated in Figure 6, powerful models like GPT-4o are highly robust to the input perturbations central to ZOO-based reprogramming. The model's textual reasoning shows it can "see past" the visual noise to the underlying image content, rendering the ZOO process that relies on these perturbations ineffective. This finding, supported by quantitative results where ZOO fails to improve over the zero-shot baseline (Table 4), requires a strategic shift away from perturbation-based adaptation for modern APIs.

In contrast, PoRL's two-stage framework is architecturally immune to this issue. By performing a one-time priming step and shifting all subsequent reprogramming to a local model, PoRL's success is not dependent on the service model's sensitivity to input noise. This allows it to achieve a remarkable **+27.8%** gain over the zero-shot baseline on GPT-4o, a task where other methods stagnate. Furthermore, on the commercial Clarifai API—a closed-vocabulary setting where zero-shot is not an option—PoRL again achieves the highest accuracy (83.2%) at a fraction of the cost ($0.20), making it over **300x cheaper** than BlackVIP ($67.30). These results confirm that PoRL's strategy provides a more robust, effective, and economically viable solution for adapting modern black-box models.

## E  USE OF LARGE LANGUAGE MODELS

We utilized a large language model as an assistive tool to polish the writing and improve the clarity of this manuscript. In addition, our experimental evaluation involves the GPT-4o model as one of the

black-box service APIs to be adapted for classification tasks. The core research ideation, experimental design, and data analysis were conducted entirely by the authors, who take full responsibility for all content presented in this paper.

# F    LIMITATIONS

Our PoRL method, while offering benefits in efficiency and local model enhancement, has certain limitations. In extreme data scarcity scenarios, such as 1/2-shot learning, the one-time priming phase may be constrained (see Fig. 3b). This can challenge the $\epsilon$-faithful primed assumption in our theoretical analysis, potentially affecting the optimality of subsequent local VR and leading to suboptimal performance. Furthermore, PoRL's performance is inherently affected by the representational capacity of the chosen local model. If the local model, even after priming, cannot capture the complexities of a particularly challenging downstream task, PoRL may not achieve the performance levels of methods that continuously leverage the full computational power and feature richness of the larger service model throughout adaptation (discussed in Sec. D.7).

