# OpenReview forum: "Prime Once, then Reprogram Locally: An Efficient Alternative to Black-Box Model Reprogramming"
_ICLR.cc/2026/Conference — ICLR 2026 Conference Withdrawn Submission_

### Official Review · Reviewer_19LN · 2025-10-31

**Soundness:** 1
**Presentation:** 3
**Contribution:** 1
**Rating:** 0
**Confidence:** 3

**Summary:**

The paper claims to present a method for black-box model reprogramming to adapt large model outputs to solve a downstream task. The method, however, boils down to training of a small network to solve the downstream task, which is initialized before training by a (some?) query (or queries?) of a large black-box model. It does not actually do any black-box model reprogramming or adaptation of the outputs of a black-box model.

**Strengths:**

The proposed approach requires orders-of-magnitude fewer API calls during training and inference, yet still outperforms other approaches (e.g ZOO methods), which don't even improve upon zero-shot performance. (but see the weakness section where I argue that this is misleading).

**Weaknesses:**

It is not clear what the blue flower block in figure 2 is supposed to represent. Is this a visual prompt? The caption should be more informative.

It is not clear where the "locally available pre-trained encoder" comes from or what its characteristics should be. What purpose is the local encoder supposed to serve here? Is it just computing useful features? If so, how does one know what would be useful for a given downstream task?

As written, the approach doesn't make sense, and appears to boil down to solving the downstream task with a simple network.
On line 084: "we query the service API just once", but on line 091 we hear that the number of API calls is 10^3. Why the discrepancy?
If the service API is only called during the priming step, then doesn't the method boil down to just training of a relatively small model (the local pre-trained encoder plus the lightweight linear layer) to solve the downstream task? All the enormous power of the service model is  only being used as a priming or initialization step for the lightweight network.
I may be missing something here, but the proposed approach seems to be just a very simple technique, where the only thing new is a method of initializing (or as the paper mentions it is similar to a distillation) the network from a large service API.

The so-called "cost-free inference" is very misleading. Of course it is cost-free - you don't use the large service model at all during inference, just the small network. It is as though we used GPT to initialize or distill AlexNet for a few iterations and then claim that we are somehow doing "black-box model reprogramming" (as the paper title suggests).

**Questions:**

See the weakness section for questions to answer.

---

### Official Review · Reviewer_H83a · 2025-11-01

**Soundness:** 2
**Presentation:** 3
**Contribution:** 1
**Rating:** 2
**Confidence:** 5

**Summary:**

Existing zeroth-order optimization approaches for adapting API-based black box models to downstream tasks are highly inefficient as they require extensive API calls. Also, robustness of modern API-based models to input perturbations makes zeroth-order optimization approaches ineffective. To address these issues, this paper proposes to simply distill the API-model to a local (pretrained encoder + linear classifier) model and then adapt the local model to the target task. In summary, the proposed approach advocates distillation followed by further adaptation.

Experiments were conducted using 10 downstream datasets with multiple teacher-student combinations and the proposed approach is shown to outperform approaches that perform black-box adaptation directly on the teacher API model.

**Strengths:**

Overall the paper is written well and was easy to follow.

Focuses on the practical problem of cost-effective adaptation of black-box API models

**Weaknesses:**

**Lack of novelty**
* Distillation is a decade old concept in ML community and distilling the predictions of a frozen teacher model to a student model is widely used. This paper is simply repackaging "distillation followed by further adaptation to target tasks" as API model adaptation.

* While the paper takes about real-world API models in the introduction to motivate the problem, the actual experiments are done using standard CLIP models and vision models. Basically, the paper is running experiments in few-shot distillation settings with standard CV models and datasets and is rebranding it. There are numerous SoTA open sourced CLIP and Vision encoder models out there today and distilling from those models would eventually lead to much better performance on the target datasets used in this paper.


**Few things in Fig.1 needs to be clarified**
* In Fig 1 (b), does #API calls include both training and inference API calls?
* Does x-axis in Fig. 1(a) represent training cost or inference cost or both?
* Which dataset is used for creating these plots?

**Questions:**

See the weaknesses mentioned above.

---

### Official Review · Reviewer_14fm · 2025-11-02

**Soundness:** 2
**Presentation:** 3
**Contribution:** 3
**Rating:** 8
**Confidence:** 2

**Summary:**

The proposed method aims to improve query efficiency of black-box adaptation approaches. Instead of repeatedly querying the API inference service for optimization, the authors leverage a locally available pretrained encoder and perform a single “priming” query with downstream task data. The resulting outputs are used to train a lightweight linear layer that aligns the local encoder’s representation space with that of the black-box model. This one-time transfer eliminates the need for further API calls, substantially reducing adaptation cost while preserving task-specific performance. The proposed one-shot priming resembles a micro-distillation step, transferring black-box model knowledge to a local encoder while minimizing API queries.

**Strengths:**

* The paper presents a clear and well-motivated objective, optimizing API-based inference efficiency. The authors empirically identify the limitations of existing approaches, providing a sound scientific rationale for their proposed technique.
* The manuscript clearly introduces the key terminologies and foundational concepts necessary to understand the contribution, ensuring accessibility for readers from related domains.
* The contribution is well-situated within the existing literature; a comparative table effectively highlights how the proposed method differs from and improves upon prior work.

**Weaknesses:**

* While the overall presentation of the paper is strong, certain sections would benefit from additional proofreading and stylistic polishing, particularly the abstract, to improve clarity and flow.
* The selection of inference services and local models evaluated appears somewhat limited. Given that the main contribution focuses on reducing API inference calls, incorporating even a small-scale evaluation using an actual proprietary API service (e.g., OpenAI’s CLIP-like endpoint or Gemini Vision) would strengthen the empirical validation and make the results more realistic and convincing.

**Questions:**

* How do you anticipate companies offering proprietary inference services will perceive the adoption of your technique? Do you foresee any potential updates to their usage policies as a response?

---

### Official Review · Reviewer_aYBg · 2025-11-06

**Soundness:** 1
**Presentation:** 3
**Contribution:** 1
**Rating:** 2
**Confidence:** 4

**Summary:**

The paper proposes a method to tackle black-box model reprogramming, which is the problem of learning a (visual) prompt to applied to input images to achieve better results on a frozen black-box model (e.g. commercial models deployed behind APIs which only allow input-output interactions). Unlike existing methods, the paper proposes a method that does not rely on Zeroth-Order Optimization (where gradients need to be estimated via changes in output losses given small perturbations of inputs), which is typically costly in terms of API calls and training time. This is done by using a local model to approximate the behavior of the black-box "service model". To accomplish this, a set of model outputs or logits are first generated by querying the local model with the training set. A linear layer is then trained on top of a pre-trained local model to adapt it to behave similarly to the service model. The local model is then prompt-tuned, and used for subsequent inference without relying on the service model. Compared to methods based on zeroth-order optimization, the paper shows that the proposed method achieves state-of-the-art performance across multiple benchmarks, while API calls are reduced by over 99%.

**Strengths:**

- The paper offers an efficient alternative to zeroth-order optimization for test-time adaptation of black-box models. This is an important and relevant problem, given the high number of API calls (shown to be on the orders of $10^8$) and long training time (shown to be over 32 hours). This makes such methods significantly more practical, and less costly, to be deployed in practice.

- The method is also shown to achieve state-of-the-art performance across a large variety of datasets.

- I think that the ability to leverage unlabeled data to achieve better alignment of the local and service model to further improve results is a strong point, since the alignment / "priming" stage simply requires distilled pseudo-labels from the service model.

**Weaknesses:**

- Upon initial review of the paper, I would expect that the method learns a visual prompt with the local model, which is then used for further inference with the *service* model. Instead, it was surprising to see that the paper proposes to discard the service model altogether, and use in its place the local fine-tuned model. This appears to defeat the entire purpose of having a powerful service model -- if one can easily distill a local model that performs better than a large commercial service model, then what is the point of having a service model in the first place?

- The paper seems to be missing a key data point in all of its experiments, which is necessary to assess the value of the proposed method:  What if one simply tunes the local model (full fine-tuning or linear layer) with the new dataset, how does it compare to using the service model for "priming" the local model?

- A use case of service models, black-box or not, is that they can be accessible for those without the necessary compute to run local models. As such training a visual prompt that can subsequently be added to all input images is attractive, since it only requires a one-time training cost, and subsequent users of the model can simply take the resulting visual prompt and append it to their queries. However the proposed method here does not train a visual prompt that works well for the service model, and instead requires all subsequent inference to be done on a local machine with the necessary compute resources to run it. This also seems to defeat the purpose of reprogramming black-box models in the first place, since it simply replaces the service black-box model with a local white-box one.

- The theoretical section appears rather self-fulfilling. In particular, to prove that the risk of the local and service model are close, the paper requires
> Assumption 2. (ϵ-Faithful Priming). The priming process is effective, resulting in the local model $F_L$ closely mimicking the logit distributions of the service model $F_S$, both immediately after priming (before VR) and after both models have been optimally reprogrammed for the downstream task.

This assumption is extremely strong, since it essentially assumes the desired conclusion that $\epsilon$ in eqn (3) is small. What should be proven and not assumed is that the "priming process is effective" for certain classes of models, under reasonable conditions. Consequently, the theory simply proves that the risk difference between the local and service model is bounded by some $\epsilon$, but says nothing about how small or large $\epsilon$ is, which makes it less meaningful.

- For the VLM experiments, the local and service models seem to be extremely similar architecture wise -- e.g. in VLM experiments, they paper uses "CLIP ViT-B/16 as the service model and ViT-B/16 as the local encoder". In the VM experiments, the service models are also architecturally very similar to the local ones (e.g. ViT-B/16 vs ViT-B/32, ResNet101 vs ResNet50), I assume likely trained on very similar, if not the exact same, datasets as well. It is not clear that this method will work on the scale of commercial API models as advertised, since those that can run locally are likely to be vastly different, in terms of both architecture and training recipe, from that offered commercially. It seems unlikely that one can be reconstructed from the other via a simple linear transformation, else this implies that the commercial model itself is not that valuable in the first place.

- I am aware of some discussion in Appendix D.8. regarding this, but do not see any details about the local model used, nor comparison tables containing results and performance of other methods.

- PoRL-MS which falls back upon expensive API zero-shot inference feels like a patch that undermines the paper's core premise, which is to achieve cost-free inference with a local model.

- As the authors mentioned, the method does not work well in extreme few shot scenarios.

**Questions:**

Please see weaknesses.

---

### Author Response · Authors · 2025-11-14
**Addressing Core Misinterpretations in Reviews for Submission 15915 [1/2]**

We thank all the reviewers for their time and valuable feedback on our submission.

While we have decided to withdraw this paper to refine its positioning, we wish to provide a response to address several key, common misunderstandings about our work's core premise, contributions, and experimental findings, as we respectfully disagree with some of the core assessments.

Our work is motivated by a critical finding: the established paradigm of Black-Box Model Reprogramming (BMR) is facing limitations. Traditional ZOO-based methods are not only prohibitively expensive but, as we demonstrate, they are becoming **ineffective on modern, robust APIs like GPT-4o** , which can simply ignore the input perturbations they rely on .

We propose PoRL (Prime Once, then Reprogram Locally) as a new, highly-efficient paradigm for the API-centric era. This strategy is explicitly designed for practical scenarios that require **fast, cost-effective, or offline adaptation** by shifting the reprogramming task to a local model after a *single*, minimal interaction with the service API.

------

We address the four most common points of confusion below.

## 1. On Novelty: "This is just distillation"

This was a key concern from Reviewers H83a, 19LN, and aYBg. We respectfully argue this misinterprets our two-stage framework and conflates our "priming" stage with traditional knowledge distillation.

Our novelty is the **"prime-then-reprogram" conceptual framework**, which is designed to solve BMR problems that standard distillation *cannot*. As we explicitly state in our paper's **"Remark (Priming vs. Knowledge Distillation)"**:

> Whereas traditional distillation aims to create a final, high-performance student model—typically requiring a shared label space—our **priming is solely a preparatory step** to make the local model more **amenable to subsequent reprogramming** . This preparatory focus allows priming to operate even when the **service and downstream label spaces are disjoint**.

This distinction is critical and directly addresses the challenges of adapting standard Vision Models (VMs) like the proprietary VM APIs in [Clarifai](https://clarifai.com/clarifai/main). For example:

- **Traditional Distillation Fails:** If the service model is a VM (e.g., pre-trained on ImageNet) and the target is Flowers102, their label spaces are disjoint. One *cannot* directly distill a "Flowers102 classifier" from an "ImageNet classifier."
- **PoRL's Priming Succeeds:** Our priming stage does not try to solve the target task. Instead, it operates *in the service model's label space*. We train the local model to match the service model's *ImageNet predictions* when given *Flowers102 images* . This preparatory step makes the local model "API-aware," and it is *this* primed model that becomes highly effective in the second *reprogramming* stage.

This is a new, "distill for reprogrammability" paradigm and the first major conceptual shift away from the limited, ZOO-based approach that has defined the BMR field.



## 2. On Missing Baselines: "You didn't compare to just training the local model"

This was a central criticism from Reviewer aYBg. We must clarify that this is **factually incorrect**, as this exact baseline was a critical part of our component analysis.

Our "Component Analysis" in **Table 6** directly compares PoRL against "Local VR" (which is precisely 'simply reprogramming the local model without priming') .

The results on EuroSAT demonstrate the value of our framework:

- **Local VR (Baseline):** 70.6% accuracy.
- **PoRL (Our Full Method):** 85.7% accuracy.

This massive **+15.1% improvement** *directly proves* that our performance gain comes from the novel "Priming" stage. This empirically validates that our method is not "just training a small network" (Reviewer 19LN) but is a synergistic framework that successfully transfers the service model's knowledge.



## 3. On Practicality: "You didn't test on real-world APIs"

This was a critique from Reviewers 14fm, H83a, and aYBg. We must point out that our paper *does* include these exact experiments in **Table 4**, and they conclusively validate our thesis.

- **Case 1: The Robust API (GPT-4o).** As we identify, modern models like GPT-4o are robust to the input noise that ZOO methods rely on. Our experiments confirm this:
  - ZOO methods (BAR, BlackVIP) fail to provide any meaningful improvement over the 59.4% zero-shot baseline.
  - PoRL achieves **87.2%** accuracy, a **+27.8% gain** where the old paradigm completely fails.
- **Case 2: The Real-world Commercial API (Clarifai).** This case provides the most practical justification for cost:
  - BlackVIP cost **$67.30** to achieve 72.1% accuracy.
  - PoRL achieved a *superior* **83.2%** accuracy for just **$0.20**.

These real-world cases demonstrate that PoRL is not just an "alternative" but is a robust, economical, and effective solution for adapting modern black-box models.

---

> ### Author Response · Authors · 2025-11-14
> **Addressing Core Misinterpretations in Reviews for Submission 15915 [2/2]**
>
> ## 4. On the Core Premise: "You 'discard' the service model" and "Cost-Free is Misleading"
>
> This was a core concern from Reviewers aYBg and 19LN, who felt that "discarding" the service model and calling the inference "cost-free" was misleading and "defeats the purpose."
>
> We respectfully disagree; this is not a weakness but is the **central design choice and main advantage of our paper**. We *intentionally* "discard" the service model at inference. This is the very innovation that enables PoRL to serve a huge range of practical, real-world scenarios where continuous, expensive API access is not feasible—such as on-device applications, edge computing, or systems requiring real-time, offline adaptation.
>
> The critique that our method is "just a small network" misses the point of BMR. The goal is to *leverage the black-box model's knowledge*. The old paradigm does this by paying for every inference. Our new paradigm does this with a one-time knowledge transfer that unlocks a local model.
>
>
>
> ## 5. On the Theoretical Framework
>
> Reviewers raised concerns that our theory was "self-fulfilling" by assuming ϵ-faithful priming and disconnected from our practical setting by assuming logit access.
>
> **A Goal, Not a "Self-Fulfilling" Assumption:** We respectfully disagree with the "self-fulfilling" characterization. The theory's assumption of $\ epsilon$-Faithful Priming is not a given; it is the *explicit goal* of our **"Prime Once"** stage. Our paper presents a *constructive framework*:
>
> 1. **We posit a principle:** *If* one can faithfully prime a local model (achieve a small ϵ), *then* a stable local optimization can provably approximate the performance of the complex black-box optimization.
> 2. **We provide the mechanism:** The "Prime Once" stage is the *practical mechanism* designed to achieve this principle.
> 3. **We show the results:** Our extensive empirical results (e.g., Tables 2, 3, 4) demonstrate that this mechanism *succeeds* in practice. The theory explains *why* this success is possible.
>
> ------
>
> We thank the reviewers again for their time. We believe our work identifies limitations in the existing BMR paradigm and provides a practical, efficient, and high-performance new path forward.

---

### Note · Authors · 2025-11-14

I have read and agree with the venue's withdrawal policy on behalf of myself and my co-authors.